# Quantum Algorithm for Online Learning of MDPs with Continuous State Space

## Abstract

We propose a novel quantum online algorithm for learning Markov Decision Processes (MDPs) with continuous state space in the average reward model. Our algorithm is based on the line of work on classical online UCCRL algorithms by Ortner and Ryabko (NeurIPS'12). To the best of our knowledge, our work is the first to consider MDPs with continuous state space in the fault-tolerant quantum setting. In the case where the state space is one-dimensional and the MDP's rewards and transition probabilities are assumed to be Lipschitz, we show that, via quantum-accessible environments, our quantum algorithm obtains a $\tilde{O}(T^{1/2})$ regret, improving upon the $\tilde{O}(T^{2/3})$ bound of Lakshmanan, Ortner, and Ryabko (PMLR'15), where $T$ is the number of iterations of the algorithm. Without the Lipschitz assumption, a regret bound of $\tilde{O}\left(T^{1/(1+\alpha)}\right)$ is obtained when $0 < \alpha < 1$ and when $\alpha \geq 1$, the regret is $\tilde{O}(\sqrt{T})$. For a general $d$-dimensional state space, the regret is bounded by $\tilde{O}(T^{1/(1+d\alpha)})$ when $d\alpha < 1$ and $\tilde{O}(\sqrt{T})$ when $d\alpha \geq 1$. Our quantum algorithm uses quantum extended value iteration as a subroutine, which is our second main contribution, and may be of independent interest. We show that quantum extended value iteration achieves a subquadratic speedup in the size of the discretized state space $\mathcal{S}$ and a quadratic speedup in the size of the action space $\mathcal{A}$, as compared to its classical counterpart. As our third contribution, we study the limiting behaviour of the sequence of value functions generated by quantum extended value iteration. We show that the sequence converges to the optimal average reward $\rho^*$ up to $\epsilon$ additive error, for some small $\epsilon > 0$.

## 1 Introduction

**Markov decision processes.** Markov Decision Processes (MDPs) [1] serve as a foundational framework for modeling decision-making in a wide array of dynamic and uncertain environments. Developed within the realm of stochastic control and mathematical optimization, MDPs provide a systematic and rigorous approach to understanding and solving sequential decision problems. An MDP models the interaction between an agent and the reinforcement learning environment. Informally, it consists of a set $\mathcal{X}$ of states, a set $\mathcal{A}$ of actions, a transition model $P$ describing the probability of moving from one state to another after taking an action and a reward function $r$. At any time step, the agent in a particular state $x^{(t)} \in \mathcal{X}$ chooses an action $a^{(t)} \in \mathcal{A}$, obtains a reward $r(x^{(t)}, a^{(t)})$, and moves to a new state $x^{(t+1)}$ according to some probability distribution $p(x^{(t+1)}|x^{(t)}, a^{(t)})$. The goal in an MDP is to find a policy $\pi$ — a mapping from states to actions — that maximizes the cumulative reward $\rho$ over time.

**Reinforcement learning.** Reinforcement learning (RL) [2] is a type of machine learning that uses MDPs as the underlying framework. In RL, an agent learns an optimal policy by interacting with an environment, receiving feedback through rewards, and using this experience to improve its decision-making. Popular RL algorithms such as Q-learning, policy gradient, value iteration, and policy iteration methods [3, 4, 5, 6, 7, 8, 9] have been widely studied. By effectively balancing between exploration and exploitation, these algorithms enhance their performance in dynamic and uncertain settings, thereby learning the optimal policy.

**Online algorithms.** Online algorithms model the interaction between an agent/learner with the environment/nature. Such algorithms are usually associated with learning or decision making [10, 11, 12, 13, 14, 15, 16, 17]. Unlike offline algorithms where the agent has full access to the training data as a whole, the setting of online algorithms is such that the agent receives part of the training data in a (possibly adversarial) sequential manner. Based on incomplete knowledge of the entire training data, the agent is required to make a decision, after which feedback from the environment is provided in the form of a gain according to a pre-defined reward function. This process is repeated for $T$ number of time steps. The maximum gain incurred when making the best fixed decision in hindsight is known as the *offline gain*. Moreover, the difference between the offline gain and the gain incurred when making some other sequence of decisions is called the *regret*. In the context of learning (communicating[1]) MDPs under the average reward model, the regret is given by $T\rho^*(M) - \sum_{t=1}^T r(x^{(t)}, a^{(t)})$, where $\rho^*(M) := \rho^*(M, x) := \max_{\pi \in \Pi} \rho_\pi(M, x)$ is the optimal average reward $\rho_\pi(x) = \frac{1}{T} \limsup_{T \to \infty} \mathbb{E}[\sum_{t=0}^T r(x^{(t)}, a^{(t)})]$ of MDP $M$ with initial state $x$ under policy $\pi$ and the maximum is taken over the set $\Pi$ of all policies [18, 19, 20]. Regret is the canonical cornerstone to benchmark the performance of an online algorithm. Typically, online algorithms with a per-step regret that scales inversely with the number of time steps $T$ are desired. This implies that given sufficiently long time, an online algorithm can perform as well as an offline algorithm.

**Our contribution.** We study the potential of quantum computing in improving the regret of online algorithm. Motivated the work of [21] which achieves an exponential improvement in learning tabular and value-target MDPs, we are interested in studying if similar improvements can be achieve in learning general MDps. We base our work on the classical framework of [18, 22]. In particular, we give a quantum version of the classical algorithm in [18, 22] and perform its regret analysis. Our contribution is threefold.

- In the average reward model, we give a quantum online algorithm that learns MDPs with continuous state space. Under the assumption that the MDPs' reward and transition probabilities are Lipschitz, our algorithm achieves a $\tilde{O}(T^{1/2})$ regret in the one-dimensional state space setting, improving upon the $\tilde{O}(T^{2/3})$ bound by [18]. Without the Lipschitz assumption, a regret bound of $\tilde{O}(T^{1/(1+\alpha)})$ is obtained when $0 < \alpha < 1$ and when $\alpha \geq 1$, the regret is $\tilde{O}(\sqrt{T})$. For a general $d$-dimensional state space, the regret is bounded by $\tilde{O}(T^{1/(1+d\alpha)})$ when $d\alpha < 1$ and $\tilde{O}(\sqrt{T})$ when $d\alpha \geq 1$.
- We propose a quantum extended value iteration subroutine. With high probability, the subroutine outputs a sequence of approximate value functions up to additive error $\epsilon$ in time $O\left(\frac{S^{1.5}\sqrt{A}}{\epsilon} \log \frac{1}{\delta}\right)$ as compared to the classical running time of $O(S^2 A)$ [19, 23, 22, 18].
- We prove convergence guarantees for an approximate analogue of value iteration to the optimal average reward $\rho^*$ up to some $\epsilon$ additive error.

**Related work.** Loosely speaking, our work is related to quantum machine learning. We discuss more details on related work in Appendix A due to space restriction. Among the most related work, Ref. [19] gave an algorithm to learn MDPs with discrete state and action spaces. Their algorithm achieves a $O(T^{1/2})$ regret, where $T$ is the number of time steps. Their work was extended to the continuous state space setting by [22], which gave a $\tilde{O}(T^{3/4})$ regret bound for one-dimensional state spaces and $\tilde{O}(T^{\frac{2d+1}{2d+2}})$ regret bound for $d$-dimensional state spaces. The followup work [18] improves upon these results, giving a regret of $\tilde{O}(T^{2/3})$ and $\tilde{O}(T^{\frac{2+d}{3+d}})$ in one- and $d$-dimensional state spaces respectively.

## 2 PRELIMINARIES

**Notations.** For any $n \in \mathbb{Z}_+$, we use $[n]$ to represent the set $\{1, \ldots, n\}$ and denote the $i$-th entry of a vector $\mathbf{v} \in \mathbb{R}^n$ by $v(i)$ for all $i \in [n]$. If a vector $\mathbf{v}$ has time dependency, we denote it as $\mathbf{v}^{(t)}$, where $t$ is the corresponding time step. The $\ell_1$ and $\ell_\infty$-norm of a vector $\mathbf{v} \in \mathbb{R}^n$ are $\|\mathbf{v}\|_1 := \sum_{i=1}^n |v(i)|$ and $\|\mathbf{v}\|_\infty := \max_{i \in [n]} |v(i)|$, respectively. We denote $\mathcal{V}$ as the space of all real-valued functions.

---

[1]The the optimal average reward $\rho^*$ does not depend on the initial state $x$.

We use $\bar{\mathbf{0}}$ to denote the all-zeros vector and $|\bar{0}\rangle$ to denote the state $|0\rangle \otimes \cdots \otimes |0\rangle$ where the number of qubits is clear from the context. We use $\mathbf{e}$ to denote the all-ones vector, $\mathbb{1}_C$ to denote the indicator function where the condition $C$ is satisfied, and $\Delta_{\mathcal{Z}}$ to denote the probability simplex on a space $\mathcal{Z}$. We use $\widetilde{O}(\cdot)$ to hide polylogarithmic factors, i.e., $\widetilde{O}(f(n)) = O(f(n) \cdot \operatorname{poly}\log(f(n)))$.

**Quantum computing**    In classical computing, the basic unit of information is a bit, which can take values 0 or 1. In quantum computing, the basic unit is known as a quantum bit, or *qubit*. It is a two-level quantum system with states $|0\rangle$ and $|1\rangle$. Unlike a classical bit that has only two states, a qubit is a *superpositions* of $|0\rangle$ and $|1\rangle$, i.e. $|v\rangle = \sum_{i=0}^{1} v_i |i\rangle$, where $v_i \in \mathbb{C}$ is the *amplitude* of $|i\rangle$ and satisfies $\sum_{i=0}^{d-1} |v_i|^2 = 1$. The states $|0\rangle, |1\rangle$ forms the (orthogonal) computational basis of the two-dimensional Hilbert space. This extends to any $d$-dimensional system, where $d > 2$. Quantum states from different Hilbert spaces can be combined using tensor product. For simplicity of notation, we use $|u\rangle |v\rangle$ to denote the tensor product $|u\rangle \otimes |v\rangle$. Operations in quantum computing are *unitary*, i.e. a linear transformation $U$ that satisfies $UU^\dagger = U^\dagger U = I$, where $U^\dagger$ is the conjugate transpose of $U$.

The information in a quantum state cannot be "read" directly. In order to observe a quantum state $|v\rangle$, we perform a *quantum measurement* on it. The measurement results in a classical state $i$ with probability $|v_i|^2$, and the measured quantum state *collapses* to $|i\rangle$. Quantum access to input data is encoded in a unitary operator known as the *quantum oracle*. Quantum oracles allow data to be accessed in superposition, thereby allowing operations to be performed "simultaneously" on states, which is the core of quantum speedups.

**Computational model.**    We refer to the running time of a quantum computation as the number of basic gates performed, excluding the gates that are used inside the oracles. We assume a quantum arithmetic model, which allows us to ignore issues arising from the fixed-point representation of real numbers. In this model, all basic arithmetic operations take constant time. We also assume a quantum circuit model, where an application of an elementary gate is equivalent to performing an elementary operation. The query complexity of a quantum algorithm with some input length is the maximum number of queries the algorithm makes on any input.

Our quantum algorithm shall commonly build KP-trees [24, 25] of vectors. In short, a KP-tree is a classical binary-tree-like data structure, with leaves storing the value of every entry of a vector and each internal node stores the sum of absolute values (or sum of absolute values squared) of its children. The root of the tree stores the $\ell_1$- (or $\ell_2$-) norm of the whole vector. For a vector $\mathbf{u} \in \mathbb{R}^S$, the KP-tree for $\mathbf{u}$ is denoted as $\mathsf{KP}_\mathbf{u}$. The KP-trees are accessible in superposition by a quantum computer via quantum random access memory (QRAM). A single query to any entry of $\mathbf{u}$ can be done in constant time. More specifically, this allows the quantum computer to query the oracles $\mathcal{O}_\mathbf{u}$ that performs the mapping $\mathcal{O}_\mathbf{u} : |s\rangle|\bar{0}\rangle \mapsto |s\rangle|u(s)\rangle \; \forall s \in \mathcal{S}$ in time $O(\operatorname{poly}\log(S))$. Moreover, (all or part of) the entries of $\mathbf{u}$ can be classically updated by writing new values into the KP-tree in at most $\tilde{O}(S)$ time.

**Quantum subroutines.**    To achieve quantum speedup and a better regret bound, we exploit a few quantum subroutines. Among them, the popular minimum finding algorithm by Dürr and Høyer [26], which can be turned straightforwardly into a maximum finding algorithm. We also use the generalized minimum finding [27] for the case when one has quantum access to the entries of $\mathbf{u}$ up to some additive error. Besides that, we use the celebrated Grover's search [28] and two other standard subroutines: quantum multi-dimensional amplitude estimation [29] and quantum multi-dimensional mean estimation [21, 30]. We restate these subroutines in Appendix B.

We tweak the standard quantum norm estimation algorithm [31, 32, 33, 34, 35] to estimate the norm of a subvector. The proof which replies on amplitude estimation and amplification [31, 36, 37, 38] is deferred to Appendix B.

**Lemma 1** (Quantum norm estimation of a subvector with additive error)**.** *Let $\delta \in (0, 1/4)$ and $\epsilon > 0$. Given a probability vector $\mathbf{p} \in [0,1]^S$ stored in $\mathsf{KP}_\mathbf{p}$, assume access to the operation $|s\rangle |0\rangle \rightarrow |s\rangle |p(s)\rangle$. Let $W \subseteq S$ be the set of entries that satisfy some given condition. Define the subvector $\mathbf{p}_W$ of $\mathbf{p}$ whose entries consist of those in $W$. There exists a quantum algorithm that*

*outputs an estimate $\tilde{\Gamma}_W$ of $\|\mathbf{p}_W\|_1 \coloneqq \sum_{s \in W} |p_W(s)|$ such that $|\tilde{\Gamma}_W - \|\mathbf{p}_W\|_1| \le \epsilon$ with success probability at least $1 - \delta$ in time $\tilde{O}(\frac{\sqrt{S}}{\epsilon} \log \frac{1}{\delta})$ and the same amount of quantum gates.*

Finally, the quantum mean estimation algorithm from [39] can be adapted to estimate the mean of a vector with real entries. The proof can be found in Appendix B.

**Lemma 2** (Quantum mean estimation). *Let $\epsilon > 0$ and $\delta \in (0, 1)$. Let $\mathbf{u} \in \mathbb{R}^S$ be a nonzero vector and $\mathbf{p} \in [0, 1]^S$ be a probability vector. Suppose we have access to $\mathsf{KP}_\mathbf{u}$, $\mathsf{KP}_\mathbf{p}$ and can make quantum queries in the form $|s\rangle |0\rangle \to |s\rangle |p(s)\rangle$ and $|s\rangle |0\rangle \to |s\rangle |u(s)\rangle$. There exists a quantum algorithm that computes an estimate $\tilde{\mu}$ of $\mu = \sum_{s \in \mathcal{S}} p(s)u(s)$ such that $|\tilde{\mu} - \mu| \le \epsilon$ with success probability at least $1 - \delta$ in time $\tilde{O}\big(\frac{\|\mathbf{u}\|_\infty}{\epsilon} \log \frac{1}{\delta}\big)$.*

## 3 MARKOV DECISION PROCESSES

A discrete-time MDP [40] can be described by a four-tuple $(\mathcal{X}, \mathcal{A}, P, r)$, where the Borel spaces $\mathcal{X}$ and $\mathcal{A}$ denote the *state* and *action* spaces, respectively. The *stochastic kernel* $P : \mathcal{X} \times \mathcal{A} \times \mathcal{X} \to [0, 1]$ is a *transition probability matrix* with entries $P(x'|x, a)$ denoting the probability to the next state $x' \in \mathcal{X}$ given that the previous state-action pair is $(x, a) \in \mathcal{X} \times \mathcal{A}$, while the *reward* function $r : \mathcal{X} \times \mathcal{A} \to \mathbb{R}$ is a measurable function. Define the history spaces $\mathcal{H}^{(0)} = \mathcal{X}$ and $\mathcal{H}^{(t)} = (\mathcal{X} \times \mathcal{A})^t \times \mathcal{X}$ for $t \in \mathbb{N}$. A policy $\pi$ is stochastic kernels on $\mathcal{A}$ given $\mathcal{X}$.[2] The set of all policies is denoted by $\Pi$.

In this work, we consider the average reward model, in which the average reward when executing policy $\pi$ with initial state $x = x^{(0)}$ is given by $\rho_\pi(x) = \limsup_{T \to \infty} \frac{1}{T} \mathbb{E}_x^\pi \big[ \sum_{t=0}^{T-1} r(x^{(t)}, a^{(t)}) \big]$, where the expectation over all $\mathcal{H}^{(\infty)} = (\mathcal{X} \times \mathcal{A})^\infty$ is taken with respect to the randomness induced by the transition probabilities and policy $\pi$. The optimal average reward $\rho^*(x)$ on initial state $x \in \mathcal{X}$ is defined as $\rho^*(x) = \sup_{\pi \in \Pi} \rho_\pi(x)$. Moreover, we say that a policy $\pi^*$ is *average optimal* if $\rho_{\pi^*}(x) = \rho^*(x)$ for all $x \in \mathcal{X}$. We assume the existence of an optimal policy $\pi^*$ with optimal average reward $\rho^*$ that is independent of the initial state. In other words, for some $\rho^* \in \mathbb{R}$, $\rho^*(x) = \rho^*$ for all $x \in \mathcal{X}$. Furthermore, we assume that for every measurable policy $\pi$, the Poisson equation $\rho_\pi + h(\pi, x) = r(x, \pi(x)) + \int p(x'|x, \pi(x))h(\pi, x')$ holds, where $h(\pi, x)$ is the *bias*[3] of policy $\pi$ in state $x$. Similar assumptions were made by [22, 18].

**Finite state approximation of MDPs.** The practical utility of MDPs lies in their ability to model decision-making in complex environments [41, 42, 43, 44, 45, 46, 47, 48, 49]. However, the computational burden associated with handling an exhaustive state and action spaces can be prohibitive. Finite state approximation addresses this challenge by allowing the system to be condensed to a more manageable and computationally tractable form, facilitating the use of various well-studied solution algorithms, such as dynamic programming and value iteration and policy iteration, which are fundamental for decision-making under uncertainty [50, 51, 52, 8, 53, 9, 54, 55].

We follow Refs. [56, 22, 57] to derive approximate MDPs with finite state and action spaces. We describe the discretization of continuous state space using $\epsilon$-nets, for some $0 < \epsilon < 1$. We make the following assumptions as in [22, 18, 57, 58]:

**Assumption 1.** *(a) $\mathcal{X}$ is compact.*

*(b) The action space $\mathcal{A}$ is finite.*

*(c) The reward $r(x, a) \in [0, 1]$ for all $x \in \mathcal{X}, a \in \mathcal{A}$.*

### 3.1 DISCRETIZATION OF STATE SPACE

Consider a continuous state space $\mathcal{X}$ with metric $d_\mathcal{X}$. By Assumption 1(a), $\mathcal{X}$ is compact and hence totally bounded. Hence, we can partition the continuous state space $\mathcal{X}$ into the finite state space $\mathcal{S} = \{s_i\}_{i=1}^S$ such that

$$\min_{s \in \mathcal{S}} d_\mathcal{X}(x, s) < \frac{1}{S} \quad \text{for all } x \in \mathcal{X}.$$

---

[2]More generally, a policy is a sequence of stochastic kernels on $\mathcal{A}$ given $\mathcal{H}^{(t)}$.

[3]The bias is the difference in accumulated rewards when starting in a different state [22].

We call $\mathcal{S}$ a $1/n$-net in $\mathcal{X}$. Define the function

$$Q_\mathcal{X} : \mathcal{X} \to \mathcal{S} \quad \text{as} \quad Q_\mathcal{X}(x) = \arg\min_{s \in \mathcal{S}} d_\mathcal{X}(x, s), \tag{1}$$

where ties are broken so that $Q_\mathcal{X}$ is measurable. The map $Q_\mathcal{X}$ is often called a nearest neighbour quantizer with respect to distortion measure $d_\mathcal{X}$ [59]. The function $Q_\mathcal{X}$ induces a partition of $\mathcal{X}$ into $\{\mathcal{X}_i\}_{i=1}^S$, where

$$\mathcal{X}_i = \{x \in \mathcal{X} : Q_\mathcal{X}(x) = s_i\} \quad \forall i \in [S].$$

For example, consider the one-dimensional setting where $\mathcal{X} = [0, 1]$. A $\frac{1}{S}$-net partitions $\mathcal{X}$ into $S$ intervals $\mathcal{X}_1, \ldots, \mathcal{X}_S$, where

$$\mathcal{X}_1 = \left[0, \frac{1}{S}\right], \quad \mathcal{X}_i = \left(\frac{i-1}{S}, \frac{i}{S}\right], \text{ for } i = 2, \ldots, S. \tag{2}$$

Each interval $\mathcal{X}_i$ is represented by a state $s_i \in \mathcal{S}$. We assume access to a discretization oracle $\mathcal{O}_\mathcal{X}$ for the state space.

**Definition 1** (Discretization oracle). *Let $\mathcal{X}$ be a state space that is continuous on $[0, 1]$ and let $\mathcal{S} \subset \mathcal{X}$ be discrete. We say that we have access to a discretization oracle $\mathcal{O}_\mathcal{X}$ if the oracle implements in constant time the mapping*

$$\mathcal{O}_\mathcal{X} : |x\rangle|\bar{0}\rangle \mapsto |x\rangle|\arg\min_{s \in \mathcal{S}} d_\mathcal{X}(x, s)\rangle \quad \forall x \in \mathcal{X}.$$

We introduce natural assumptions for rewards and transition probabilities in nearby states. Similar assumptions have been considered in [56, 22, 57].

**Assumption 2.** *For any $x, x', \in \mathcal{X}$ and any $a, a' \in \mathcal{A}$, there exists a constant $L > 0$ such that*

$$|r(x, a) - r(x', a)| \le L |x - x'|^\alpha, \tag{3}$$

$$\|\mathbf{p}(\cdot|x, a) - \mathbf{p}(\cdot|x', a)\|_1 \le L |x - x'|^\alpha \tag{4}$$

Under Assumption 2, the bias of the optimal policy is bounded [22, 18]. We assume that $L$ from Eqs. (3) and (4) are the same. Similar assumptions were also made by [22, 18, 19].

Here, we clarify some notations that will be used in the remaining parts of the paper. We use the subscript $\pi$ to denote MDP parameters induced by the policy $\pi$. For a discrete state space $\mathcal{S}$ and action space $\mathcal{A}$ with cardinalities $S$ and $A$ respectively, define the reward vector $\mathbf{r}_\pi \in [0, 1]^S$ and transition probability matrix $\mathbf{P}_\pi \in [0, 1]^{S \times S}$ as

$$r_\pi(s) = \mathbb{E}_{a \sim \pi(\cdot|s)}[r(s, a)], \quad p_\pi(s'|s) = \mathbb{E}_{a \sim \pi(\cdot|s)}[p(s'|s, a)].$$

## 4 VALUE ITERATION

Value iteration [1, 7] is a dynamic programming algorithm used to find the optimal policy for a reinforcement learning algorithm. The goal is to determine the best action to be taken in each state in order to maximize its cumulative expected rewards. Value iteration has been widely used and exists in different variants [53, 60, 61, 62, 63, 64, 65]. The algorithm updates the value function $\mathbf{u} \in \mathbb{R}^S$ of all states $s \in \mathcal{S}$ according to the update rule[4]

$$\mathbf{u}^{(0)} = \mathbf{0}, \qquad \mathbf{u}^{(i+1)} = \max_{\pi \in \Pi} \left\{ \mathbf{r}_\pi + \mathbf{P}_\pi \mathbf{u}^{(i)} \right\}, \tag{5}$$

and has a per-iteration running time of $O(S^2 A)$ [66].

---

[4]We use $\max_{\pi \in \Pi}\{\cdot\}$ and $\max_{a \in \mathcal{A}}\{\cdot\}$ interchangeably in this paper since we will be considering the greedy policy approach throughout this work.

## 4.1 APPROXIMATE VALUE ITERATION

Approximate value iteration has been well-studied and is used in various settings [67, 68, 69, 70, 71, 72]. From here on, we shall refer to value iteration with update rule (5) as *standard* value iteration. In this section, we consider an approximate analogue of the value iteration algorithm which differs from the standard value iteration algorithm in the following ways:

1. Denote the value function output by approximate value iteration as $\tilde{\mathbf{u}}$. In standard value iteration, $\mathbf{P}_\pi \mathbf{u}$ in Eq. (5) is computed exactly, while in approximate value iteration, it is estimated up to an additive error. In particular, let $\tilde{\boldsymbol{\mu}}_\pi$ denote the estimate of $\mathbf{P}_\pi \tilde{\mathbf{u}}$ such that

$$\|\tilde{\boldsymbol{\mu}}_\pi - \mathbf{P}_\pi \tilde{\mathbf{u}}\|_\infty \leq \frac{\epsilon}{2}. \tag{6}$$

2. The maximization in Eq. (5) is computed exactly in standard value iteration. However, in approximate value iteration, given Eq. (6), the maximization is estimated up to additive error $\epsilon$. In order words, define the operator $\mathcal{L}' : \mathbb{R}^S \to \mathbb{R}^S$ on $\tilde{\mathbf{u}}$, then $\mathcal{L}' \tilde{\mathbf{u}} \geq \max_{\pi \in \Pi}\{\mathbf{r}_\pi + \tilde{\mathbf{P}}_\pi \tilde{\mathbf{u}}\} - \epsilon \mathbf{e}$.

For any integer $i \geq 0$, a single run of the approximate value iteration recursion can be expressed as

$$\tilde{\mathbf{u}}^{(i+1)} = \mathcal{L}' \tilde{\mathbf{u}}^{(i)}. \tag{7}$$

In this section, we show the convergence of approximate value iteration. First, we need the following claim whose proof is deferred to Appendix C.1. Let us for now consider non-communicating MDPs, whose optimal average reward $\rho^*(s)$ is dependent on the initial state $s$.

**Claim 1.** *Let $\epsilon \in (0, 1)$ and fix $i \in \mathbb{Z}_{\geq 0}$. Let $\mathbf{u}^{(i+1)} \in \mathbb{R}^S$ be the value function obtained after $i$ steps of standard value iteration and let $\tilde{\mathbf{u}}^{(i+1)} \in \mathbb{R}^S$ be its corresponding approximation obtained after $i$ steps of approximate value iteration. Then*

$$\mathbf{u}^{(i+1)} - (i+1)\epsilon \mathbf{e} \leq \tilde{\mathbf{u}}^{(i+1)} \leq \mathbf{u}^{(i+1)}.$$

The theorem below, whose proof is moved to Appendix C.1, illustrates the limiting behaviour of the sequence of value functions output by approximate value iteration.

**Theorem 1.** *Let $\epsilon \in (0, 1)$. For all $\mathbf{u}^{(0)} \in \mathcal{V}$ and all $s \in \mathcal{S}$,*

$$\boldsymbol{\rho}^* - \epsilon \mathbf{e} \leq \liminf_{i \to \infty} \frac{\tilde{\mathbf{u}}^{(i)}}{i} \leq \limsup_{i \to \infty} \frac{\tilde{\mathbf{u}}^{(i)}}{i} \leq \boldsymbol{\rho}^*.$$

Theorem 1 implies the following corollary (proof in Appendix C.1).

**Corollary 1.** *Let $\epsilon \in (0, 1)$ and let $\pi$ be a policy such that $\pi^\infty = (\pi, \pi, \cdots)$ is average optimal. Theorem 1 implies that*

$$\|\boldsymbol{\rho}^* - \mathbf{P}_\pi \boldsymbol{\rho}^*\|_\infty \leq \epsilon.$$

We say that a policy $\bar{\pi}$ is $u$-improving if $\bar{\pi} \in \arg\max_{\pi \in \Pi}\{\mathbf{r}_\pi + \mathbf{P}_\pi \mathbf{u}\}$. The next theorem bounds the optimal reward (see Appendix C). The proof can be found in Appendix C.1.

**Theorem 2.** *Let $\pi$ be any $\mathbf{u}$-improving policy and $\rho^* \in \mathbb{R}$ be the optimal average reward. Let $\mathcal{L}'$ be a single run of approximate value iteration. Then, the following holds for all $s \in \mathcal{S}$:*

$$\min_{s \in \mathcal{S}}\{\mathcal{L}' u(s) - u(s)\} \leq \rho^{\pi^\infty} \leq \rho^* \leq \max_{s \in \mathcal{S}}\{\mathcal{L}' u(s) - u(s)\} + \epsilon. \tag{8}$$

## 4.2 EXTENDED VALUE ITERATION

Consider the set $\mathcal{M}$ of all MDPs with common state space $\mathcal{S}$, common action space $\mathcal{A}$, transition probabilities $\tilde{p}(\cdot|s, a)$ and mean rewards $\tilde{r}(s, a)$ such that

$$\|\tilde{\mathbf{p}}(\cdot|s, a) - \hat{\mathbf{p}}(\cdot|s, a)\|_1 \leq d(s, a) \tag{9}$$

$$|\tilde{r}(s, a) - \hat{r}(s, a)| \leq d'(s, a) \tag{10}$$

for some given probability distributions $\hat{\mathbf{p}}(\cdot|s, a)$, given rewards $\hat{r}(s, a) \in [0, 1], d(s, a) > 0$, and $d'(s, a) \geq 0$. Furthermore, assume that $\mathcal{M}$ contains at least one communicating[5] MDP. Extended value iteration updates the value function of all $s \in \mathcal{S}$ of $\mathcal{M}$ [22, 18, 19, 1] using the rule

$$u^{(0)}(s) = 0; \quad u^{(i+1)}(s) = \max_{a \in \mathcal{A}} \left\{ \tilde{r}(s, a) + \max_{\mathbf{p}(\cdot) \in \mathcal{P}(s,a)} \left\{ \sum_{s' \in \mathcal{S}} p(s') \cdot u^{(i)}(s') \right\} \right\}, \quad (11)$$

where $\tilde{r}(s, a) = \hat{r}(s, a) + d'(s, a)$ are the maximal possible rewards according to Eq. (10) and $\mathcal{P}(s, a)$ is the set of transition probabilities $\tilde{\mathbf{p}}(\cdot|s, a)$ satisfying Eq. (9). The classical algorithm by [23, Proposition 2] finds the solution $\mu_{\max}(s, a)$ to the inner maximization problem $\max_{\mathbf{p}(\cdot) \in \mathcal{P}(s,a)} \left\{ \sum_{s' \in \mathcal{S}} p(s') \cdot v^{(i)}(s') \right\}$ of Eq. (11) in $O(S)$ time. In addition, solving the other maximization takes $O(A)$ time. This leads to a per-iteration run time of $O(S^2 A)$ to update the values $u^{(i+1)}(s)$ for all $s \in \mathcal{S}$.

We propose a quantum algorithm that improves upon the per-iteration run time of extended value iteration by a subquadratic factor in $S$ and a quadratic factor in $A$. Specifically, we give a quantum subroutine that outputs an estimate $\tilde{\mu}_{\max}(s, a)$ of $\mu_{\max}(s, a)$ up to additive accuracy $\epsilon$ with success probability at least $1 - \delta$ in time $\tilde{O}\left( \frac{\sqrt{S}}{\epsilon} \log \frac{1}{\delta} \right)$.

**Lemma 3.** *Let $\epsilon, \delta \in (0, 1)$ and $u_{\min}, u_{\max} \in \mathbb{R}$. Consider the set $\mathcal{P}(s, a)$ of transition probabilities that satisfy Eq. (9). Let $\hat{\mathbf{p}} \in [0, 1]^S$ be a transition probability vector such that $\hat{\mathbf{p}} \in \mathcal{P}(s, a)$ and let $\mathbf{u} \in [u_{\min}, u_{\max}]^S$ be a nonzero vector. Given quantum access to the entries of $\hat{\mathbf{p}}, \mathbf{u}$ that are stored in KP-trees $\mathsf{KP}_{\hat{\mathbf{p}}}$ and $\mathsf{KP}_{\mathbf{u}}$ respectively, there exists a quantum algorithm that outputs an estimate $\tilde{\mu}$ of $\mu^* = \max_{\mathbf{p}(\cdot) \in \mathcal{P}(s,a)} \sum_{s' \in \mathcal{S}} p(s') \cdot u(s')$ such that $|\tilde{\mu} - \mu^*| \leq \epsilon$ with success probability at least $1 - \delta$. The time complexity is $\tilde{O}\left( \frac{\sqrt{S}}{\epsilon} \log \frac{1}{\delta} \right)$.*

Using Lemma 3, we present the following result.

**Lemma 4** (Guarantees of one iteration of quantum extended value iteration). *Let $\epsilon, \delta \in (0, 1)$. Fix $i \in \mathbb{Z}_{\geq 0}$. Given access to estimated rewards $\hat{r}(s, a)$, estimated maximum mean value $\tilde{\mu}_{max}(s, a)$ and distance $d(s, a)$ for a state-action pair, there exists a quantum algorithm that outputs an estimate $\tilde{u}^{(i+1)}(s)$ of the solution $u^{(i+1)}(s)$ to Eq. (11) such that $\tilde{u}^{(i+1)}(s) \geq u^{(i+1)}(s) - \epsilon$ with success probability at least $1 - \delta$ for all $s \in \mathcal{S}$. This requires $\tilde{O}\left( \frac{S^{1.5}\sqrt{A}}{\epsilon} \log \frac{1}{\delta} \right)$ time.*

The pseudocodes and proofs of Lemmas 3 and 4 can be found in Appendix C.2. Next, we prove the convergence of quantum extended value iteration (proof in Appendix C.2).

**Theorem 3** (Convergence of quantum extended value iteration). *Let $\epsilon, \delta \in (0, 1)$. Let $\mathcal{M}$ be the set of all MDPs with state space $\mathcal{S}$, action space $\mathcal{A}$, transition probabilities $\tilde{\mathbf{p}}(\cdot|s, a)$, and mean rewards $\tilde{r}(s, a)$ that satisfy Eqs. (9) and (10) for given probability distributions $\hat{\mathbf{p}}(\cdot|s, a)$, values $\hat{r}(s, a) \in [0, 1], d(s, a) > 0$, and $d'(s, a) \geq 0$. If $\mathcal{M}$ contains at least one communicating MDP, quantum extended value iteration (Algorithm 3, see Appendix ) satisfies*

$$\boldsymbol{\rho}^* - \epsilon\mathbf{e} \leq \lim_{i \to \infty} \frac{\tilde{\mathbf{u}}^{(i)}}{i} \leq \boldsymbol{\rho}^*.$$

*Furthermore, terminating quantum extended value iteration (Algorithm 3) when*

$$\max_{s \in \mathcal{S}} \left\{ \tilde{u}^{(i+1)}(s) - \tilde{u}^{(i)}(s) \right\} - \min_{s \in \mathcal{S}} \left\{ \tilde{u}^{(i+1)}(s) - \tilde{u}^{(i)}(s) \right\} \leq \epsilon,$$

*the greedy policy with respect to $\tilde{\mathbf{u}}^{(i)}$ is $\epsilon$-optimal.*

# 5 QUANTUM ALGORITHM FOR ONLINE LEARNING MDPS

## 5.1 QUANTUM-ACCESSIBLE ENVIRONMENTS

Classically, we are able to directly observe complete trajectories $(s^{(0)}, a^{(0)}, s^{(1)}, a^{(1)}, \dots)$ in every episode and collect samples to estimate $\hat{r}(s, a)$ and $\hat{\mathbf{p}}(\cdot|s, a)$ for any $(s, a) \in \mathcal{S} \times \mathcal{A}$ [18, 22, 19]. In

---

[5]We say that an MDP is communicating if for every pair of states $s, s'$ in $\mathcal{S}$, there exists a deterministic stationary policy $\pi^\infty$ under which $s'$ is accessible from $s$.

the quantum setting, we can only collect quantum states via quantum-accessible environments. This has been studied by [73, 74, 75, 21]. The following oracles are required.

**Definition 2** (Quantum sampling oracle for transition probabilities [76]). *Let $\mathcal{X}$ be a continuous state space and $\mathcal{S}$ be the resulting state space after discretization. For any $s \in \mathcal{S}$ and $a \in \mathcal{A}$, a quantum sampling oracle for transition probabilities $\mathcal{O}_p$ performs the following mapping:*

$$\mathcal{O}_p : |s\rangle \, |a\rangle \, |\bar{0}\rangle \rightarrow \int_{x \in \mathcal{X}} \sqrt{p(x|s,a)dx} \, |s\rangle \, |a\rangle \, |x\rangle \otimes |garbage(x)\rangle \,, \tag{12}$$

*where the second quantum register denotes possible garbage quantum states that arise in the implementation of the oracle. We let $\mathcal{O}_{p^{(t)}}$ denote the quantum sampling oracle for transition probabilities at step $t \in \mathbb{Z}_+$ on inputs $s^{(t)}, a^{(t)}$.*

**Definition 3** (Quantum reward oracle). *Let $\mathcal{S}$ and $\mathcal{A}$ be discrete state and action spaces respectively. For any $s \in \mathcal{S}$ and $a \in \mathcal{A}$, a quantum reward oracle $\mathcal{O}_r$ performs the mapping: $\mathcal{O}_r : |s\rangle \, |a\rangle \, |\bar{0}\rangle \rightarrow |s\rangle \, |a\rangle \, |r(s,a)\rangle$.*

**Definition 4** (Quantum policy oracle). *Let $\mathcal{S}$ and $\mathcal{A}$ be discrete state and action spaces respectively. For any $s \in \mathcal{S}$ and $a \in \mathcal{A}$, we say that we have access to a quantum policy oracle $\mathcal{O}_\pi$ that does the mapping $\mathcal{O}_\pi : |s\rangle \, |\bar{0}\rangle \rightarrow \sum_{a \in \mathcal{A}} \sqrt{\pi(a|s)} \, |s\rangle \, |\pi(s)\rangle$.*

In this work, we use the Classical Sampling via Quantum Access (CSQA) [21] procedure (see Algorithm 4 in Appendix D) to simulate the classical sampling of a state $s^{(t)} \sim d_\pi^{(t)}$ when given a policy $\pi$ and time step $t$, where $d_\pi^{(t)}$ is the probability distribution over $\mathcal{S}$ according to policy $\pi$ and at time step $t$. We show the following lemma whose proof is in Appendix D.

**Lemma 5.** *Given a policy $\pi$ and an integer $t \in \mathbb{Z}_+$. Let $d_\pi^{(t)}$ be the probability distribution over states $s \in \mathcal{S}$ at step $t$ according to $\pi$. Suppose we have access to oracle $\mathcal{O}_\mathbf{p}$ (see Definition 2 in Appendix D), then there exists a quantum algorithm that outputs a sample of $s \sim d_\pi^{(t)}$ in time $O(t)$.*

We present our quantum algorithm for online learning MDPs in Algorithm 1. Our quantum algorithm implements "optimism in the face of uncertainty". It maintains a set of plausible MDPs $\mathcal{M}$ and optimistically chooses an MDP $\tilde{M} \in \mathcal{M}$ and a policy $\tilde{\pi}$ such that the average reward $\rho_{\tilde{\pi}}(\tilde{M})$ is maximized up to $\frac{\epsilon}{\sqrt{T}}$ error, for $T$ number of iterations of the algorithm. Similar to Ref. [22], we assume an MDP to be plausible if its aggregated rewards and transition probabilities are within a certain range (see Eqs. (13) and (14)).

The corresponding estimated rewards and transition probabilities are computed from sampled values of action $a$ in the state close to $x$. Specifically, the state space is partitioned into $\mathcal{X}_1, \cdots, \mathcal{X}_S$ as a result of discretization. The corresponding aggregated transition probabilities are defined as

$$p^{\mathrm{agg}}(\mathcal{X}_j|x,a) \coloneqq \int_{\mathcal{X}_j} p(dx'|x,a).$$

In this work, we write $\mathbf{p}^{\mathrm{agg}}(\cdot)$ to denote the aggregated probability distribution with respect to $\{\mathcal{X}_1, \ldots, \mathcal{X}_S\}$ for a probability distribution $\mathbf{p}(\cdot)$ over $\mathcal{X}$. Given the aggregated state space $\{\mathcal{X}_1, \ldots, \mathcal{X}_S\}$, estimates $\hat{r}(x,a)$ and $\hat{\mathbf{p}}^{\mathrm{agg}}(\cdot|x,a)$ are obtained from all samples of action $a$ in states $x \in \mathcal{X}$ represented by $s \in \mathcal{S}$ after discretization. As a consequence, the estimates are the same for states $x \in \mathcal{X}$ represented by the same $s \in \mathcal{S}$.

As in the UCCRL algorithms in Refs. [18, 22, 19], our algorithm proceeds in episodes, in which the chosen policy remains fixed. The algorithm moves to a new episode when the number of visitations to a state-action pair has been doubled, after which the estimates of rewards and transition probabilities are updated. Furthermore, since all states $x$ represented by the same $s$ have the same confidence interval, finding the optimal pair $\tilde{M}_k, \tilde{\pi}_k$ in Eq. (15) is equivalent to finding the optimistic discretized MDP $\tilde{M}_k^{\mathrm{agg}}$ and an optimal policy $\tilde{\pi}_k^{\mathrm{agg}}$ on $\tilde{M}_k^{\mathrm{agg}}$. Hence, $\tilde{\pi}_k$ can be viewed as the extension of $\tilde{\pi}_k^{\mathrm{agg}}$ to $\mathcal{X}$. In other words, $\tilde{\pi}_k(x) \coloneqq \tilde{\pi}_k^{\mathrm{agg}}(s)$, where $s \in \mathcal{S}$ is the state representing the interval $\mathcal{X}_j$ that $x$ belongs to, for some $j \in [n]$ [22].

**Algorithm 1** Quantum algorithm for online learning MDPs

**Input:** State space $\mathcal{X}$, action space $\mathcal{A}$, confidence parameter $\delta$, upper bound $H$ on the bias span, Lipschitz parameters $L$.

1: Define $\mathcal{X}_j$ as in Eq. (2), where each interval $\mathcal{X}_j$ is represented by a state $s_j$ for all $j \in [S]$.
2: Set $t = 1$.
3: Initialize $\hat{\mathbf{p}}_1(\cdot|s, a) = (1/S, \cdots, 1/S) \in \mathbb{R}^S$ and $\hat{r}_1(s, a) = 0.5$ for all $s \in \mathcal{S}, a \in \mathcal{A}$.
4: **for** episodes $k = 1, 2, \cdots$ **do**
5:      Let $N_k(s, a) =$ be the number of times action $a$ has been chosen in a state in the interval represented by $s$, prior to episode $k$ and let $n_k(s, a)$ be the respective counts in episode $k$.
6:      Set the start time of episode $k$, $t_k := t$.
7:      **for** $(s, a) \in \mathcal{S} \times \mathcal{A}$ **do**
8:          Initialize $v_k(s, a) = 0$.
9:      **end for**
10:     Let $\mathcal{M}_k$ be the set of plausible MDPs $\tilde{M}$ with $H(\tilde{M}) \leq H$ and rewards $\tilde{r}(x, a)$ and transition probabilities $\tilde{\mathbf{p}}(\cdot|x, a)$ such that

$$|\tilde{r}(x, a) - \hat{r}_k(x, a)| \leq LS^{-\alpha} + \frac{\sqrt{SA}}{\max\{1, N_k(s, a)\}} \tag{13}$$

and

$$\|\tilde{\mathbf{p}}^{\mathrm{agg}}(\cdot|x, a) - \hat{\mathbf{p}}_k^{\mathrm{agg}}(\cdot|x, a)\|_1 \leq LS^{-\alpha} + \frac{S}{\max\{1, N_k(s, a)\}} \tag{14}$$

11:     Choose policy $\tilde{\pi}_k'$ and $\tilde{M}' \in \mathcal{M}_k$ such that

$$\rho_{\tilde{\pi}_k}\left(\tilde{M}_k\right) \geq \arg\max\{\rho^*(M)|M \in \mathcal{M}_k\} - \frac{\epsilon}{\sqrt{T}} \tag{15}$$

using Algorithm 3.
12:     **while** $n_k\left(s^{(t)}, a^{(t)}\right) < \max\left\{1, N_k\left(s^{(t)}, a^{(t)}\right)\right\}$ **do**
13:         Call $x^{(t)} := CSQA(\tilde{\pi}_k', t)$ using Algorithm 4, query $\mathcal{O}_{\mathcal{X}}$ on $x^{(t)}$ to obtain $s^{(t)}$ and let $a^{(t)} := \tilde{\pi}_k'\left(x^{(t)}\right)$.
14:         Update $n_k\left(s^{(t)}, a^{(t)}\right) = n_k\left(s^{(t)}, a^{(t)}\right) + 1$.
15:         Set $t = t + 1$.
16:     **end while**
17:     **for** $s \in \mathcal{S}$ and $a \in \mathcal{A}$ **do**
18:         Compute estimate $\hat{r}_k(x, a)$ up to additive error $\frac{\sqrt{SA}}{\max\{1, N_k(s, a)\}}$ with probability at least $1 - \frac{\delta}{24T^{5/4}}$ using Fact 4 and by invoking oracles $\mathcal{O}_{\mathcal{X}}, \mathcal{O}_p, \mathcal{O}_r$.
19:         Compute estimate $\hat{\mathbf{p}}_k^{\mathrm{agg}}(\cdot|x, a)$ up to additive error $\frac{S}{\max\{1, N_k(s, a)\}}$ in the $\ell_1$-norm with probability at least $1 - \frac{\delta}{24T^{5/4}}$ using Fact 4 invoking oracles $\mathcal{O}_{\mathcal{X}}, \mathcal{O}_p, \mathcal{O}_r$.
20:     **end for**
21: **end for**

The theorem below states that a $\tilde{O}(1/\sqrt{T})$ regret bound is attainable by Algorithm 1. The proof is deferred to Appendix E.

**Theorem 4.** *Let $M$ be an MDP with continuous state space $[0, 1]$, $A$ actions, rewards and transition probabilities satisfying Eqs. (3) and (4), and bias span at most $H$. Then, the regret of Algorithm 1 after $T$ steps is upper bounded by*

$$2(H + 1)LTS^{-\alpha} + (14 + 15H)SA \log \frac{SAT}{\delta} + (2H + 3)\sqrt{T} \log \frac{SAT}{\delta}$$

*with probability at least $1 - \delta$. Furthermore, setting $S = T^{\frac{1}{1+\alpha}}$ gives a regret bound of*

$$(2H + 1)LT^{\frac{1}{1+\alpha}} + (14 + 15H)AT^{\frac{1}{1+\alpha}} \log \frac{AT^2}{\delta} + (2H + 3)\sqrt{T} \log \frac{AT^2}{\delta}.$$

*Taking $H = \log T$, we obtain a regret bound of $\tilde{O}(\sqrt{T})$ when $\alpha \geq 1$ and $\tilde{O}(T^{\frac{1}{1+\alpha}})$ when $0 < \alpha < 1$.*

## 6 DISCUSSION AND CONCLUSION

We study the problem of online learning MDPs with continuous state space. In this setting, only the state and action spaces are known to the algorithm. Other parameters of the MDPs such as the reward function and transition probabilities are unknown.

We give a quantum algorithm, that in each episode, chooses an optimistic MDP and its corresponding (nearly) optimal policy. This is done using a quantum subroutine, called quantum extended value iteration. The chosen policy is then executed until some action in some state-actions pair has been visited as often in the episode as before the episode. The observed rewards are accumulated and the regret is analyzed. Our results show that the quantum algorithm achieves a $\tilde{O}(\sqrt{T})$ regret when the state space is one-dimensional and assuming that the MDPs' rewardds and transition probabilities are Lipschitz. This improves upon the regret bound obtained by [18]. Without the Lipschitz assumption, the regret is $\tilde{O}(T^{1/(1+\alpha)})$ when $0 < \alpha < 1$ and $\tilde{O}(\sqrt{T})$ when $\alpha \geq 1$. This bound also implies that MDPs with continuous state space can be learned with the same (order in $T$) regret as those with discrete state space when $\alpha \geq 1$. For the case where the state space is $d$-dimensional ($d \geq 1$), the regret is bounded by $\tilde{O}(T^{1/(1+d\alpha)})$ when $d\alpha < 1$ and $\tilde{O}(\sqrt{T})$ when $d\alpha \geq 1$.

We point out that a similar work to ours has been done by Ref. [21]. Unlike our quantum algorithm that learns general MDPs, the quantum algorithm proposed in [21] learns specific MDPs, i.e. tabular and value target MDPs. Furthermore, episodes in the algorithm of [21] have fixed length. This allows their algorithm to achieve a logarithmic regret in $T$, the number of episodes. This is in contrast to our algorithm, whose length of episodes grows indefinitely with $T$.

The quantum extended value iteration subroutine is a combination of techniques such as quantum mean estimation, quantum norm estimation and quantum minimum finding with approximate unitary. It has a per-iteration runtime of $O\left(\frac{S^{1.5}\sqrt{A}}{\epsilon} \log \frac{1}{\delta}\right)$, achieving a speedup that is subquadratic in the size of the discretized state space $S$ and quadratic in the size of the action space $A$, as compared to its classical counterpart. By studying the limiting behaviour of the sequence of value functions $\{\tilde{\mathbf{u}}^{(i)}\}$ generated by an approximate analogue of standard value iteration, we show that quantum extended value iteration converges up to additive error $\epsilon$ and the greedy policy with respect to the value function is $\epsilon$-optimal. Furthermore, the sequence $\{\tilde{\mathbf{u}}^{(i)}\}$ when compared to that generated by standard value iteration $\{\mathbf{u}^{(i)}\}$, satisfies $\mathbf{u}^{(i)} - i\epsilon\mathbf{e} \leq \tilde{\mathbf{u}}^{(i)} \leq \mathbf{u}^{(i)}$ for some $\epsilon > 0$ and any $i \geq 1$. We hope that our quantum extended value iteration algorithm and its analysis would be of independent interest to readers.

We highlight some future directions following our work. In this work, we follow the approach of [22, 18, 57] to discretize the state space using $\epsilon$-nets. It would be interesting to learn if other discretization methods could lead to better regret bounds of the algorithm. Besides, the lower bound on the regret still remains an open problem since the work of [18, 22]. Other future directions include extending the problem setting to continuous action space.

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

## A  RELATED WORK

In the discounted reward model, Bas-Serrano *et al.* [77] proposed a logistic Q-learning algorithm, which is closely related to the relative entropy policy search algorithm [78]. Using a convex loss function for policy evaluation, the algorithm outputs a sequence of policies whose average quality approaches that of the optimal policy. Subsequently, Neu and Olkhovskaya [20] incorporated the algorithm of [77] into their online algorithm to learn MDPs with linear function approximation in the setting where the reward function is allowed to change adversarily between episodes, obtaining a $\tilde{O}(\sqrt{T})$ regret, where $T$ denotes the number of episodes.

In the average reward model, Meyn [79] studied policy iteration in general (continuous) state spaces. Their algorithm was shown to output a sequence of policies that satisfy a strong stability condition and finds an optimal average cost policy under further conditions. Besides that, Ref. [58], under MDPs with continuous state and action spaces, presented an approximate relative value iteration algorithm that outputs a sequence of piecewise-linear convex relative value functions, which has a monotonically non-decreasing lower bound on the average reward. Another work that considers MDPs with continuous state and action spaces is Ref. [57], which gave a discretization-based approximation method for MDPs with continuous spaces, accompanied by a detailed error analysis. They also developed synchronous and asynchronous Q-learning algorithms for continuous spaces via discretization. In the online learning framework, Auer, Jaksch, and Ortner [19] gave an algorithm to learn MDPs with discrete state and action spaces. Their algorithm achieves a $O(\sqrt{T})$ regret, where $T$ is the number of time steps. Their work was extended to the continuous setting by Ortner and Ryabko [22], who gave a $\tilde{O}(T^{3/4})$ regret bound for 1-dimensional state space and $\tilde{O}(T^{\frac{2d+1}{2d+2}})$ when the state space is $d$-dimensional. The follow-up work of Lakshmanan, Ortner, and Ryabko [18] improved upon these results, giving a regret of $\tilde{O}(T^{2/3})$ and $\tilde{O}(T^{\frac{2+d}{3+d}})$ in 1 and $d$-dimensional state space, respectively.

In the quantum setting, Wiedemann *et al.* [73] gave a full implementation and simulation of a policy iteration algorithm that is based on amplitude amplification. Besides numerically showing that the policy output by their algorithm is close to optimal, they conjectured that a quadratic speedup in the size of the set of all possible policies as compared to classical Monte Carlo estimation methods is achievable. Wang *et al.* [74] gave two quantum algorithms that approximate an optimal policy, the optimal value function, and the optimal Q-function using quantum mean estimation and quantum maximum finding. They showed a quadratic improvement over the best possible classical sample complexities with respect to the approximation error, the effective time horizon, and the size of the action space. On the other hand, two quantum policy gradient algorithms were developed by Jerbi *et al.* [75] to estimate the optimal policy using quantum numerical and analytical gradient estimation respectively, gaining a quadratic reduction in sample complexity over their classical analogues when the trained policies satisfy certain conditions. Based on the classical least-squares policy iteration algorithm [8], Cherrat *et al.* [80] gave a general framework for quantum reinforcement learning via policy iteration using block-encoding techniques [81, 82]. They showed that the value functions output by the algorithms in their framework are close to optimal. Finally, the first line of study on exploration in online quantum reinforcement learning was done by Zhong *et al.* [21] who showed a worst-case regret guarantee that scales logarithmically in the number of episodes, beating the $\Omega(\sqrt{T})$ regret lower bound in classical reinforcement learning. Subsequently, Ganguly *et al.* [83] gave an upper-confidence-bound-based quantum algorithm that achieves an exponential improvement in regret and quadratic improvement in the sample complexity as compared to the classical counterparts. The aforementioned works consider the discounted reward model and MDPs with discrete state and action spaces. Other related work in the near-term regime includes [84, 85, 86, 87, 88].

## B  QUANTUM SUBROUTINES

In this section, we restate the quantum subroutines that we use in our paper, starting with quantum minimum finding by Dürr and Høyer [26].

**Fact 1** (Quantum minimum finding [26])**.** *Given quantum access to a vector* $\mathbf{u} \in \mathbb{R}^n$*, we can find* $u_{\min} := \min_{i \in [n]} u(i)$ *with success probability* $1 - \delta$ *using* $O(\sqrt{n} \log \frac{1}{\delta})$ *queries and* $\tilde{O}(\sqrt{n} \log \frac{1}{\delta})$ *quantum gates.*

The above minimum finding algorithm can be turned straightforwardly into a maximum finding algorithm. The quantum minimum finding algorithm was later generalized by Chen and de Wolf [27] for the case when one has quantum access to the entries of $u$ up to some additive error.

**Fact 2** (Quantum min-finding with an approximate unitary [27]). *Let $\delta_1, \delta_2 \in (0,1)$ such that $\delta_2 = O\big(\delta_1^2/(S\log(1/\delta_1))\big)$, $\epsilon > 0$, and $\mathbf{u} \in \mathbb{R}^S$. Suppose access to a unitary that maps $|s\rangle |\bar{0}\rangle \mapsto |s\rangle |\Lambda(s)\rangle$ such that, for every $s \in [S]$, after measuring the state $|\Lambda(s)\rangle$, with probability at least $1 - \delta_2$ the first register $|\tilde{u}(s)\rangle$ of the measurement outcome satisfies $|\tilde{u}(s) - u(s)| \leq \epsilon$. Then there is a quantum algorithm that finds an index $s$ such that $u(s) \leq \min_{s' \in \mathcal{S}} u(s') + 2\epsilon$ with probability at least $1 - \delta_1$ and in time $\widetilde{O}(\sqrt{S}\log(1/\delta_1))$.*

The next result is the celebrated Grover's quantum search algorithm.

**Fact 3** (Grover's search [28]). *Let $m, n \in \mathbb{Z}_+$ such that $m < n/2$. Given quantum access to an unsorted database of $n$ elements with $m$ marked elements, there exists a quantum algorithm that finds a marked element in $O(\sqrt{\frac{n}{m}})$ time.*

The next two results are essential to obtain a better regret bound.

**Fact 4** (Quantum multidimensional amplitude estimation [21, 29]). *Let $\epsilon, \delta \in (0,1)$. Assume access to a probability oracle $U_\mathbf{p} : |0\rangle \to \sum_{i=1}^n \sqrt{p(i)} |i\rangle |\psi_i\rangle$ for any $n$-dimensional probability distribution $p$ and ancillary quantum sates $\{|\psi_i\rangle\}_{i=1}^n$. There exists a quantum algorithm that returns an approximation $\tilde{\mathbf{p}}$ of $\mathbf{p}$ such that $\|\tilde{\mathbf{p}} - \mathbf{p}\|_1 \leq \epsilon$ with success probability at least $1 - \delta$ using $O\big(\frac{n}{\epsilon}\log\frac{n}{\delta}\big)$ quantum queries to $U_p$ and its inverse.*

**Fact 5** (Quantum multidimensional mean estimation [21, 30]). *Let $\epsilon, \delta \in (0,1)$. Let $X : \Omega \to \mathbb{R}^n$ be an $n$-dimensional bounded variable on a probability space $(\Omega, \mathbf{p})$ such that $\|X\|_2 \leq C$ for some constant $C$. Assume access to the probability oracle $U_\mathbf{p} : |0\rangle \to \sum_{\omega \in \Omega} \sqrt{p(\omega)} |\omega\rangle |\phi_\omega\rangle$ for ancillary quantum states $\{|\phi_\omega\rangle\}_{\omega \in \Omega}$ and a binary oracle $U_X : |\omega\rangle |0\rangle \to |\omega\rangle |X(\omega)\rangle$ for all $\omega \in \Omega$. Then there is a quantum algorithm that outputs an estimate $\tilde{\boldsymbol{\mu}}$ of $\boldsymbol{\mu} = \mathbb{E}[\mathbf{X}]$ such that $\|\tilde{\boldsymbol{\mu}}\|_2 \leq C$ and $\|\tilde{\boldsymbol{\mu}} - \boldsymbol{\mu}\|_\infty \leq \epsilon$ with success probability at least $1 - \delta$ using $O\big(\frac{C}{\epsilon}\log\frac{n}{\delta}\big)$ quantum queries to $U_\mathbf{p}, U_\mathbf{X}$ and their inverses.*

We tweak the standard quantum norm estimation algorithm [31, 32, 33, 34, 35] to estimate the norm of a subvector.

**Lemma 1** (Quantum norm estimation of a subvector with additive error). *Let $\delta \in (0, 1/4)$ and $\epsilon_1 \in (0, S]$. Given a probability vector $\mathbf{p} \in [0,1]^S$ stored in $\mathsf{KP_p}$, assume access to the operation $|s\rangle |0\rangle \to |s\rangle |p(s)\rangle$. Let $W \subseteq [S]$ and define the subvector $\mathbf{p}_W$ of $\mathbf{p}$ whose entries consist of those in $W$. There exists a quantum algorithm that outputs an estimate $\tilde{\Gamma}_W$ of $\|\mathbf{p}_W\|_1 \coloneqq \sum_{s \in W} |p_W(s)|$ such that $|\tilde{\Gamma}_W - \|\mathbf{p}_W\|_1| \leq \epsilon$ with success probability at least $1 - \delta$ in time $\tilde{O}(\frac{\sqrt{S}}{\epsilon}\log\frac{1}{\delta})$ and using the same amount of $\tilde{O}(\frac{\sqrt{S}}{\epsilon}\log\frac{1}{\delta})$ quantum gates.*

*Proof.* Using query access to the probability vector $\mathbf{p}$, create a circuit to prepare the state $\frac{1}{\sqrt{S}}\sum_{s \in \mathcal{S}} |s\rangle |p(s)\rangle |0\rangle$. Define a *good-states*-controlled rotation as

$$U_{good} |p(s)\rangle |0\rangle = \begin{cases} |p(s)\rangle \left(\sqrt{p(s)} |1\rangle + \sqrt{1 - p(s)} |0\rangle\right) & \text{if } s \in W, \\ |p(s)\rangle |0\rangle & \text{if } s \notin W. \end{cases} \tag{16}$$

Perform the controlled-rotation in Eq. (16) to get the state

$$\frac{1}{\sqrt{S}}\sum_{s \in W} |s\rangle |p(s)\rangle \left(\sqrt{p(s)} |1\rangle + \sqrt{1 - p(s)} |0\rangle\right) + \frac{1}{\sqrt{S}}\sum_{s \notin W} |s\rangle |p(s)\rangle |0\rangle$$

$$= \frac{1}{\sqrt{S}}\sum_{s \in W} \sqrt{p(s)} |s\rangle |p(s)\rangle |1\rangle + \left(\frac{1}{\sqrt{S}}\sum_{s \in W} \sqrt{1 - p(s)} |s\rangle |p(s)\rangle + \frac{1}{\sqrt{S}}\sum_{s \notin W} |s\rangle |p(s)\rangle\right) |0\rangle$$

$$= \sqrt{a} |\phi_1\rangle |1\rangle + \sqrt{1 - a} |\phi_0\rangle |0\rangle \tag{17}$$

for some normalized states $|\phi_0\rangle, |\phi_1\rangle$, where $a = \sum_{s \in W} \frac{p(s)}{S} = \frac{\|\mathbf{p}_W\|_1}{S}$.

Let $U_u$ be the unitary that prepares the state in Eq. (17) and define the new unitaries $U = U_u(I - 2|\bar{0}\rangle\langle\bar{0}|)U_u^\dagger$ and $V = I - I \otimes |1\rangle\langle 1|$. Nondestructive unbiased amplitude estimation [36, 37, 38] allows us to obtain an estimate $\tilde{a}$ of $a = \frac{\|\mathbf{p}_W\|_1}{S}$ such that $|\mathbb{E}[\tilde{a}] - a| \le \frac{\epsilon_0^2}{32}$ and $\mathrm{Var}(\tilde{a}) \le \frac{91a}{K^2} + \frac{\epsilon_0^2}{32}$, restoring the initial state with success probability at least $1 - \frac{\epsilon_0^2}{32}$, using $O\left(K \log\log K \log(K/\epsilon_0)\right)$ expected number of applications of $U$ and $V$. Setting $K > \frac{8}{\epsilon_0}\sqrt{91a}$ via exponential search without knowledge of $a$, we have

$$\mathbb{P}\left[|\tilde{a} - \mathbb{E}[\tilde{a}]| \ge \frac{\epsilon_0}{2}\right] \le \frac{4}{\epsilon_0^2}\left(\frac{91a}{K^2} + \frac{\epsilon_0^2}{32}\right) \le \frac{4}{\epsilon_0^2}\left(\frac{\epsilon_0^2}{64} + \frac{\epsilon_0^2}{32}\right) \le \frac{1}{16} + \frac{1}{8} \le \frac{1}{4}.$$

by Chebyshev's inequality. The success probability $3/4$ is boosted with $O(\log\frac{1}{\delta})$ repetitions to $1 - \delta/2$ via the median of means technique. Hence,

$$|\tilde{a} - a| \le |\tilde{a} - \mathbb{E}[\tilde{a}]| + |\mathbb{E}[\tilde{a}] - a| \le \epsilon_0/2 + \epsilon_0/2 = \epsilon_0$$

with success probability at least $1 - 4\delta$. The quantity $\tilde{\Gamma}_W := S\tilde{a}$ is thus an estimate

$$|\tilde{\Gamma}_W - \|\mathbf{p}_W\|_1| = S|\tilde{a} - a| \le S\epsilon_0 = \epsilon.$$

We set $\epsilon_0 = \epsilon/S$, which means that setting $K > \frac{8S}{\epsilon}\sqrt{91a} = \frac{8}{\epsilon_1}\sqrt{91\|\mathbf{p}_W\|_1 S}$ is sufficient. This brings the total runtime to $O\left(\frac{\sqrt{S}}{\epsilon}\log\log\frac{\sqrt{S}}{\epsilon}\log\frac{S^{3/2}}{\epsilon^2}\log\frac{1}{\delta}\right)$ in expectation. While Ref. [38] proved a result in expected time, we use the probabilistic result obtained from Markov's inequality and repetition at a cost of another factor of $O(\log\frac{1}{\delta})$. $\qquad\square$

Lastly, we adapt the quantum mean estimation algorithm from [39] to estimate the mean of a vector with real entries.

**Lemma 2** (Quantum mean estimation). *Let $\epsilon > 0$ and $\delta \in (0, 1/8)$. Let $\mathbf{u} \in \mathbb{R}^S$ be a nonzero vector and $\mathbf{p} \in [0,1]^S$ be a probability vector. Suppose we have access to $\mathsf{KP_p}, \mathsf{KP_u}$ and can make quantum queries in the form $|s\rangle|a\rangle|s'\rangle|0\rangle \to |s\rangle|a\rangle|s'\rangle|p(s'|s,a)\rangle$ and $|s\rangle|0\rangle \to |s\rangle|u(s)\rangle$. There exists a quantum algorithm that computes an estimate $\tilde{\mu}$ of $\mu = \sum_{s'\in\mathcal{S}} p(s') \cdot u(s')$ such that $|\tilde{\mu} - \mu| \le \epsilon$ with success probability at least $1 - 9\delta$ in time $\tilde{O}\left(\frac{\|\mathbf{u}\|_\infty}{\epsilon}\log\frac{1}{\delta}\right)$.*

*Proof.* Prepare the state $\sum_{s'\in\mathcal{S}}\sqrt{p(s'|s,a)}|s'\rangle|\bar{0}\rangle$ using $O(\log n)$ queries to $\mathcal{O}_{\mathsf{KP_p}}, \mathcal{O}_{\mathsf{KP_p}}^\dagger$ and elementary gates. Query $\mathcal{O}_{\mathsf{KP_u}}$ to obtain

$$\sum_{s'\in\mathcal{S}}\sqrt{p(s'|s,a)}|s'\rangle|u(s')\rangle|0\rangle. \tag{18}$$

Throughout this proof, we will use $p(s')$ to denote $p(s'|s,a)$ for brevity. Define the positive-controlled rotation such that

$$U_{CR^+}: |a\rangle|0\rangle \to \begin{cases} |a\rangle\left(\sqrt{a}|1\rangle + \sqrt{1-a}|a\rangle\right) & \text{if } a \in [0,1], \\ |a\rangle|0\rangle & \text{otherwise.} \end{cases}$$

Apply $U_{CR^+}$ on the last two registers in Eq. (18). Using quantum maximum finding to find $\|\mathbf{u}\|_\infty$ with success probability $1 - \delta$, we obtain

$$|\psi\rangle = \sum_{s'\in\mathcal{S}:u(s')>0}\sqrt{p(s')}|s'\rangle|u(s')\rangle\left(\sqrt{\frac{u(s')}{\|\mathbf{u}\|_\infty}}|1\rangle + \sqrt{1 - \frac{u(s')}{\|\mathbf{u}\|_\infty}}|0\rangle\right) + \sum_{s'\in\mathcal{S}:u(s')\le 0}p(s')|s'\rangle|u(s')\rangle|0\rangle$$

$$= \sum_{s'\in\mathcal{S}:u(s')>0}\sqrt{\frac{p(s')\cdot u(s')}{\|\mathbf{u}\|_\infty}}|s'\rangle|u(s')\rangle|1\rangle$$

$$+ \left(\sum_{s'\in\mathcal{S}:u(s')>0}\sqrt{p(s') - \frac{p(s')\cdot u(s')}{\|\mathbf{u}\|_\infty}}|s'\rangle|u(s')\rangle + \sum_{s'\in\mathcal{S}:u(s')\le 0}\sqrt{p(s')}|s'\rangle|u(s')\rangle\right)$$

$$= \sqrt{\mu_+}|\phi_1\rangle|1\rangle + \sqrt{1-\mu_+}|\phi_0\rangle|0\rangle,$$

where $\mu_+ = \sum_{s' \in \mathcal{S}: u(s') > 0} \frac{p(s') \cdot u(s')}{\|\mathbf{u}\|_\infty}$.

Let $U_u$ be the unitary that creates the state $|\psi\rangle$ and define the new unitaries $U = U_u(I - 2|\bar{0}\rangle\langle\bar{0}|)U_u^\dagger$ and $V = I - I \otimes |1\rangle\langle 1|$. Nondestructive unbiased amplitude estimation [36, 37, 38] allows us to obtain an estimate $\tilde{a}$ of $a = \sum_{s' \in \mathcal{S}: u(s') > 0} \frac{p(s') \cdot u(s')}{\|\mathbf{u}\|_\infty}$ such that $|\mathbb{E}[\tilde{a}] - a| \leq \frac{\epsilon_0^2}{128}$ and $\mathrm{Var}(\tilde{a}) \leq \frac{91a}{K^2} + \frac{\epsilon_0^2}{128}$, restoring the initial state with success probability at least $1 - \frac{\epsilon_0^2}{128}$, using $O\left(K \log\log K \log(K/\epsilon_0)\right)$ expected number of applications of $U$ and $V$. Setting $K > \frac{16}{\epsilon_0}\sqrt{91a}$ via exponential search without knowledge of $a$, we have

$$\mathbb{P}\left[|\tilde{a} - \mathbb{E}[\tilde{a}]| \geq \frac{\epsilon_0}{4}\right] \leq \frac{16}{\epsilon_0^2}\left(\frac{91a}{K^2} + \frac{\epsilon_0^2}{128}\right) \leq \frac{16}{\epsilon_0^2}\left(\frac{\epsilon_0^2}{256} + \frac{\epsilon_0^2}{128}\right) \leq \frac{1}{16} + \frac{1}{8} \leq \frac{1}{4}$$

by Chebyshev's inequality. The success probability $3/4$ is boosted with $O(\log\frac{1}{\delta})$ repetitions to $1 - \delta/2$ via the median of means technique. Hence,

$$|\tilde{a} - a| \leq |\tilde{a} - \mathbb{E}[\tilde{a}]| + |\mathbb{E}[\tilde{a}] - a| \leq \epsilon_0/4 + \epsilon_0/4 = \epsilon_0/2$$

with success probability at least $1 - 4\delta$. Hence the quantity $\tilde{\mu}_+ := \|\mathbf{u}\|_\infty \tilde{a}$ is an estimate

$$\left|\tilde{\mu}_+ - \sum_{s' \in \mathcal{S}: u(s') > 0} p(s') \cdot u(s')\right| \leq \|\mathbf{u}\|_\infty |\tilde{a} - a| \leq \|\mathbf{u}\|_\infty \epsilon_0/2 = \epsilon/2.$$

We set $\epsilon_0 = \epsilon/\|\mathbf{u}\|_\infty$, which means that setting $K > \frac{16\|\mathbf{u}\|_\infty}{\epsilon}\sqrt{91a}$ is sufficient. This brings the total run time to $O\left(\frac{\|\mathbf{u}\|_\infty}{\epsilon} \log\log \frac{\|\mathbf{u}\|_\infty}{\epsilon} \log \frac{\|\mathbf{u}\|_\infty}{\epsilon^2} \log\frac{1}{\delta}\right)$ in expectation. While Ref. [38] proved a result in expected time, we use the probabilistic result obtained from Markov's inequality and repetition at a cost of another factor of $O(\log\frac{1}{\delta})$.

We similarly compute the estimate $\tilde{\mu}_-$ of

$$\mu_- = \sum_{s' \in \mathcal{S}: u(s') \leq 0} \frac{p(s') \cdot u(s')}{\|\mathbf{u}\|_\infty}$$

up to additive error $\frac{\epsilon}{2}$ with success probability at least $1 - 4\delta$. Now, notice that

$$\mu = \sum_{s' \in \mathcal{S}} p(s') \cdot u(s') = \sum_{s' \in \mathcal{S}: u(s') > 0} \frac{p(s') \cdot u(s')}{\|\mathbf{u}\|_\infty} - \sum_{s' \in \mathcal{S}: u(s') \leq 0} \frac{p(s') \cdot u(s')}{\|\mathbf{u}\|_\infty} = \mu_+ - \mu_-.$$

Let $\tilde{\mu} = \tilde{\mu}_+ - \tilde{\mu}_-$. Hence, we obtain

$$|\tilde{\mu} - \mu| = |(\tilde{\mu}_+ - \mu_+) - (\tilde{\mu}_- - \mu_-)| \leq |\tilde{\mu}_+ - \mu_+| + |\tilde{\mu}_- - \mu_-| \leq \epsilon$$

with success probability at least $1 - 9\delta$. $\qquad\square$

## C  VALUE ITERATION

Below, we review some useful facts on the convergence of value iteration which we will use to prove the convergence of approximate value iteration in the next subsection. In particular, these results revolve around the limiting behaviour of the sequence $\{e^{(i)}\}$, where

$$\mathbf{e}^{(i)} \equiv \mathbf{u}^{(i)} - i\boldsymbol{\rho}^* - \mathbf{h}^*.$$

We use the operator $\mathcal{L} : \mathbb{R}^S \to \mathbb{R}^S$ to denote a single run of value iteration, i.e. $\mathbf{u}^{(i+1)} = \mathcal{L}\mathbf{u}^{(i)}$ for any $i \in \mathbb{Z}_+$. We start with a result on the bounds on the optimal reward and on the optimality of the policy derived from standard value iteration. We say that a policy $\bar{\pi}$ is $u$-improving if $\bar{\pi} \in \arg\max_{\pi \in \Pi}\{\mathbf{r}_\pi + \mathbf{P}_\pi\mathbf{u}\}$.

**Fact 6** ([1], Theorem 8.5.5). *Let $\mathcal{L}$ be defined as above, $\rho^*$ be the optimal average reward and $\rho^{\pi^\infty}$ be average reward obtained by a deterministic stationary policy $\pi^\infty$. Then, for all $s \in \mathcal{S}$, any $\mathbf{u}^{(0)} \in \mathcal{V}$ and any $\mathbf{u}$-improving policy $\pi$,*

$$\min_{s \in \mathcal{S}}\{\mathcal{L}u(s) - u(s)\} \leq \rho^{\pi^\infty} \leq \rho^* \leq \max_{s \in \mathcal{S}}\left[\mathcal{L}u(s) - u(s)\right].$$

The following result bounds the value of $\mathbf{e}^{(i)}$ and shows that $\frac{\mathbf{u}^{(i)}}{i}$ converges to the optimal reward $\boldsymbol{\rho}^*$ as $i \to \infty$.

**Fact 7** ([1, Theorem 9.4.1(b)]). *For all $\mathbf{u}^{(0)} \in \mathcal{V}$,*

$$\min_{i \to \infty} \frac{\mathbf{u}^{(i)}}{i} = \boldsymbol{\rho}^*.$$

### C.1 RESULTS ON APPROXIMATE VALUE ITERATION

In this subsection, we restate our results on the limiting behaviour and performance of approximate value iteration, together with their proofs.

**Claim 1.** *Let $\epsilon \in (0, 1)$ and fix $i \in \mathbb{Z}_{\geq 0}$. Let $\mathbf{u}^{(i+1)}$ be the value function obtained after $i$ steps of standard value iteration and let $\tilde{\mathbf{u}}^{(i+1)}$ be its corresponding approximation obtained after $i$ steps of approximate value iteration. Then*

$$\mathbf{u}^{(i+1)} - (i + 1)\epsilon \mathbf{e} \leq \tilde{\mathbf{u}}^{(i+1)} \leq \mathbf{u}^{(i+1)}.$$

*Proof.* We prove the left-hand side of the inequality by induction. As the base case when $i = 1$, we have

$$\begin{aligned}
\tilde{\mathbf{u}}^{(1)} &\geq \max_{a \in \mathcal{A}} \left\{ \mathbf{r}_\pi + \tilde{\boldsymbol{\mu}}_\pi \right\} - \frac{\epsilon}{2} \mathbf{e} \\
&\geq \max_{a \in \mathcal{A}} \left\{ \mathbf{r}_\pi + \mathbf{P}_\pi \tilde{\mathbf{u}}^{(0)} - \frac{\epsilon}{2} \mathbf{e} \right\} - \frac{\epsilon}{2} \mathbf{e} \\
&= \max_{a \in \mathcal{A}} \left\{ \mathbf{r}_\pi + \mathbf{P}_\pi \mathbf{u}^{(0)} \right\} - \mathbf{e} \\
&= \mathbf{u}^{(1)} - \mathbf{e}.
\end{aligned}$$

Suppose that the induction hypothesis is true for all $i = k$. Then when $i = k + 1$,

$$\begin{aligned}
\tilde{\mathbf{u}}^{(k+1)} &\geq \max_{a \in \mathcal{A}} \left\{ \mathbf{r}_\pi + \boldsymbol{\mu}_\pi^{(k)} \right\} - \frac{\epsilon}{2} \mathbf{e} \\
&\geq \max_{a \in \mathcal{A}} \left\{ \mathbf{r}_\pi + \mathbf{P} \tilde{\mathbf{u}}^{(k)} - \frac{\epsilon}{2} \mathbf{e} \right\} - \frac{\epsilon}{2} \mathbf{e} \\
&\geq \max_{a \in \mathcal{A}} \left\{ \mathbf{r}_\pi + \mathbf{P} \mathbf{u}^{(k)} - k\epsilon \mathbf{e} - \frac{\epsilon}{2} \mathbf{e} \right\} - \frac{\epsilon}{2} \mathbf{e} \\
&= \mathbf{u}^{(k+1)} - (k + 1)\epsilon \mathbf{e}.
\end{aligned}$$

The right-hand side of the inequality is due to the fact that the actions chosen in approximate value iteration are at most as good as the ones chosen in standard value iteration, resulting in a $\tilde{\mathbf{u}}^{(i)}$ value that is at most $\mathbf{u}^{(i)}$. This completes the proof. $\square$

**Theorem 1.** *Let $\epsilon \in (0, 1)$. For all $\mathbf{u}^{(0)} \in \mathcal{V}$ and all $s \in \mathcal{S}$,*

$$\boldsymbol{\rho}^* - \epsilon \mathbf{e} \leq \liminf_{i \to \infty} \frac{\tilde{\mathbf{u}}^{(i)}}{i} \leq \limsup_{i \to \infty} \frac{\tilde{\mathbf{u}}^{(i)}}{i} \leq \boldsymbol{\rho}^*.$$

*Proof.* By Claim 1, we have

$$\mathbf{u}^{(i)} - i\epsilon \mathbf{e} \leq \tilde{\mathbf{u}}^{(i)} \leq \mathbf{u}^{(i)}.$$

Dividing throughout by $i$ and taking the limit as $i \to \infty$ gives

$$\lim_{i \to \infty} \frac{\mathbf{u}^{(i)}}{i} - \epsilon \mathbf{e} \leq \liminf_{i \to \infty} \frac{\tilde{\mathbf{u}}^{(i)}}{i} \leq \limsup_{i \to \infty} \frac{\tilde{\mathbf{u}}^{(i)}}{i} \leq \lim_{i \to \infty} \frac{\mathbf{u}^{(i)}}{i}.$$

By Fact 7, we get

$$\boldsymbol{\rho}^* - \epsilon \mathbf{e} \leq \liminf_{i \to \infty} \frac{\tilde{\mathbf{u}}^{(i)}}{i} \leq \limsup_{i \to \infty} \frac{\tilde{\mathbf{u}}^{(i)}}{i} \leq \boldsymbol{\rho}^*.$$

This completes the proof. $\square$

Theorem 1 implies the following corollary

**Corollary 1.** *Let $\epsilon \in (0,1)$ and let $\pi$ be a policy such that $\pi^\infty$ is average optimal. Theorem 1 implies that*

$$\|\boldsymbol{\rho}^* - \mathbf{P}_\pi \boldsymbol{\rho}^*\|_\infty \leq \epsilon.$$

*Proof.* By Claim 1 and by definition of $\mathbf{u}^{(i+1)}$,

$$\mathbf{r}_\pi + \mathbf{P}_\pi \mathbf{u}^{(i)} - (i+1)\epsilon\mathbf{e} \leq \tilde{\mathbf{u}}^{(i+1)} \leq \mathbf{u}^{(i+1)}.$$

Dividing throughout by $i+1$ and taking the limit as $i \to \infty$, we obtain

$$\lim_{i \to \infty} \frac{\mathbf{r}_\pi}{i+1} + \mathbf{P}_\pi \left( \lim_{i \to \infty} \frac{\mathbf{u}^{(i)}}{i+1} \right) - \lim_{i \to \infty} \frac{(i+1)\epsilon}{i+1}\mathbf{e} \leq \lim_{i \to \infty} \frac{\tilde{\mathbf{u}}^{(i+1)}}{i+1} \leq \lim_{i \to \infty} \frac{\mathbf{u}^{(i+1)}}{i+1}.$$

Using Theorem 1 and Fact 7, we have

$$\mathbf{P}_\pi \boldsymbol{\rho}^* - \epsilon\mathbf{e} \leq \liminf_{i \to \infty} \frac{\tilde{\mathbf{u}}^{(i+1)}}{i+1} \leq \limsup_{i \to \infty} \frac{\tilde{\mathbf{u}}^{(i+1)}}{i+1} \leq \boldsymbol{\rho}^*,$$

which is equivalent to

$$\|\mathbf{P}_\pi \boldsymbol{\rho}^* - \boldsymbol{\rho}^*\|_\infty \leq \epsilon. \qquad \square$$

We say that a policy $\bar{\pi}$ is $u$-improving if $\bar{\pi} \in \arg\max_{\pi \in \Pi}\{\mathbf{r}_\pi + \mathbf{P}_\pi \mathbf{u}\}$. The next theorem bounds the optimal reward for every state $s \in \mathcal{S}$.

**Theorem 2.** *Let $\pi$ be any $\mathbf{u}$-improving policy and $\rho^* \in \mathbb{R}$ be the optimal average reward. Let $\mathcal{L}'$ be a single run of approximate value iteration. Then, the following holds for all $s \in \mathcal{S}$:*

$$\min_{s \in \mathcal{S}} \{\mathcal{L}'u(s) - u(s)\} \leq \rho^{\pi^\infty} \leq \rho^* \leq \max_{s \in \mathcal{S}} \{\mathcal{L}'u(s) - u(s)\} + \epsilon. \tag{19}$$

*Proof.* Let $\mathcal{L}$ be a single run of standard value iteration. By construction of $\mathcal{L}$ and $\mathcal{L}'$, for all $s \in \mathcal{S}$,

$$\mathcal{L}u(s) - u(s) - \epsilon \leq \mathcal{L}'u(s) - u(s) \leq \mathcal{L}u(s) - u(s),$$

which implies that

$$\min_{s \in \mathcal{S}} \{\mathcal{L}'u(s) - u(s)\} \leq \min_{s \in \mathcal{S}} \{\mathcal{L}u(s) - u(s)\}$$

and

$$\max_{s \in \mathcal{S}} \{\mathcal{L}u(s) - u(s)\} \leq \max_{s \in \mathcal{S}} \{\mathcal{L}'u(s) - u(s) + \epsilon\}.$$

Given Fact 6, we conclude that for all $s \in \mathcal{S}$ and any $\mathbf{u}$-improving policy $\pi$,

$$\min_{s \in \mathcal{S}} \{\mathcal{L}'u(s) - u(s)\} \leq \rho^{\pi^\infty} \leq \rho^* \leq \max_{s \in \mathcal{S}} \{\mathcal{L}'u(s) - u(s)\} + \epsilon. \qquad \square$$

## C.2 EXTENDED VALUE ITERATION

The classical algorithm by [23, Proposition 2] finds the solution $\mu_{\max}(s,a)$ to the inner maximization problem $\max_{p(\cdot) \in \mathcal{P}(s,a)} \left\{ \sum_{s' \in \mathcal{S}} p(s') \cdot v^{(i)}(s') \right\}$ of Eq. (11) in $O(S)$ time. The approach is to place as much transition probability as possible on the state with the largest value $u(s)$ at the expense of transition probabilities on states with small $u(s)$. In particular, they first sort the states according to their values $u(s)$. This takes $O(S)$ time. Then, for the state $s_{\max}$ that has the highest value $u(s_{\max})$, set $p(s_{\max}) = \hat{p}(s_{\max}|s,a) + \frac{d(s,a)}{2}$. For the remaining states, set $p(s') = \hat{p}(s'|s,a)$. Note that $p$ is no longer a probability distribution since $\sum_{s' \in \mathcal{S}} p(s') = 1 + \frac{d(s,a)}{2}$. The vector $p$ is then truncated on entries that correspond to states with the smallest values $u(s)$. In particular, an iterative procedure of setting the entry of $p$ that corresponds to the states with the smallest value $u(s)$ to $p(s) = \max\left\{0, 1 - \sum_{s' \neq s} p(s')\right\}$ is carried out. This takes $O(S)$ time.

## C.3 QUANTUM EXTENDED VALUE ITERATION

We give a quantum algorithm for extended value iteration. Specifically, we first give a quantum subroutine that outputs an estimate $\tilde{\mu}_{\max}(s, a)$ of $\mu_{\max}(s, a)$ up to additive accuracy $\epsilon$ with success probability at least $1 - \delta$ in time $\tilde{O}\left(\frac{\sqrt{S}}{\epsilon} \log \frac{1}{\delta}\right)$. The approach is similar to that of [19, 23]. In particular, we find the state $s_{\max}$ and set $p(s_{\max}) = \hat{p}(s_{\max}) + \frac{d(s,a)}{2}$. For the remaining states $s' \in \mathcal{S}\backslash\{s_{\max}\}$, we set $p(s') = \hat{p}(s'|s, a)$. For the truncation step, we search over the values $u(s')$ for a cut-off point $c$. We call all states $s' \in \mathcal{S}$ with value $u(s') \leq c$ *good states with respect to* $c$. We require that the transition probabilities of good states with respect to $c$ satisfy

$$\sum_{s' \in \mathcal{S}: u(s') \leq c} \hat{p}(s'|s, a) - \frac{d(s, a)}{2} \leq \epsilon_{\text{gap}},$$

for some small $\epsilon_{\text{gap}} \in (0, 1)$. In order to find the cut-off point, we perform a binary search over the values $u(s')$ of all $s' \in \mathcal{S}$. At every iteration of the search, we perform $\ell_1$-norm estimation on the vector

$$\hat{p}_{\text{good}}(s') = \begin{cases} \hat{p}(s'|s, a) & \text{if } s' \text{ is a good state with respect to } c, \\ 0 & \text{otherwise,} \end{cases}$$

up to additive error $\epsilon_{\text{norm}}$. After $\tilde{O}(\log S)$ iterations, binary search converges to an estimated cut-off point $\tilde{c}$. We set $p(s') = 0$ for all the good states with respect to $\tilde{c}$.

We describe the steps to compute $\tilde{\mu}_{\max}(s, a)$ in Algorithm 2, while the lemma below discusses the guarantee of Algorithm 2.

**Lemma 3.** *Let $\epsilon, \delta \in (0, 1)$ and $u_{\min}, u_{\max} \in \mathbb{R}$. Consider the set $\mathcal{P}(s, a)$ of transition probabilities that satisfy Eq. (9). Let $\hat{\mathbf{p}} \in [0, 1]^S$ be a transition probability vector such that $\hat{\mathbf{p}} \in \mathcal{P}$ and let $\mathbf{u} \in [u_{\min}, u_{\max}]^S$ be a nonzero vector. Given quantum access to the entries of $\hat{\mathbf{p}}, \mathbf{u}$ that are stored in KP-trees $\mathsf{KP}_{\hat{\mathbf{p}}}$ and $\mathsf{KP}_{\mathbf{u}}$ respectively, there exists a quantum algorithm that outputs an estimate $\tilde{\mu}$ of $\mu^* = \max_{p(\cdot) \in \mathcal{P}(s,a)} \sum_{s' \in \mathcal{S}} p(s') \cdot u(s')$ such that $|\tilde{\mu} - \mu^*| \leq \epsilon$ with success probability at least $1 - \delta$. The time complexity is $\tilde{O}\left(\frac{\sqrt{S}}{\epsilon} \log \frac{1}{\delta}\right)$.*

*Proof.* First, we show that binary search eventually terminates. In particular, we prove that the search range decreases in every step. Let the search range for iteration $t$ be $[\text{low}^{(t)}, \text{high}^{(t)}]$. There are three cases:

- If $\left|\tilde{\Gamma}_{\leq c}^{(t)} - \frac{d(s,a)}{2}\right| \leq \epsilon_{\text{gap}}$, the algorithm returns $c^{(t)}$ and we are done.

- If $\tilde{\Gamma}_{<c^{(t)}} \geq \frac{d(s,a)}{2} + \epsilon_{\text{gap}}$, then the new search range will be updated to $[\text{low}^{(t)}, c^{(t)}]$. We see that

$$\text{high}^{(t+1)} - \text{low}^{(t+1)} = c^{(t)} - \text{low}^{(t)} = \frac{\text{high}^{(t)} - \text{low}^{(t)}}{2} - \text{low}^{(t)} = \frac{\text{high}^{(t)} - \text{low}^{(t)}}{2} \leq \text{high}^{(t)} - \text{low}^{(t)}$$

- If $\tilde{\Gamma}_{\leq c} \leq \frac{d(s,a)}{2} - \epsilon_{\text{gap}}$, then the new search range will be updated to $[c^{(t)}, \text{high}^{(t)}]$. We see that

$$\text{high}^{(t+1)} - \text{low}^{(t+1)} = \text{high}^{(t)} - c^{(t)} = \text{high}^{(t)} - \frac{\text{high}^{(t)} - \text{low}^{(t)}}{2} = \frac{\text{high}^{(t)} - \text{low}^{(t)}}{2} \leq \text{high}^{(t)} - \text{low}^{(t)}$$

Eventually, the condition $\text{high}^{(t)} - \text{low}^{(t)} \leq \epsilon_{\text{dist}}$ is met. Next, we show that the following are equivalent:

(a) $\left|c^{(t+1)} - c^{(t)}\right| \leq \frac{\epsilon_{\text{dist}}}{2}$;

(b) $\left|\text{high}^{(t)} - \text{low}^{(t)}\right| \leq \epsilon_{\text{dist}}$

---

**Algorithm 2** Quantum algorithm to compute the inner maximization problem of Eq. (11)

---

**Input:** Quantum access to estimates $\hat{\mathbf{p}}(\cdot|s,a)$ stored in $\mathsf{KP}_{\hat{\mathbf{p}}}$ and $\mathsf{KP}_{\mathbf{u}}$, and distance $d(s,a)$ for a state-action pair $(s,a)$, failure probability $\delta \in (0,1)$, errors $\epsilon_{\text{mean}}, \epsilon'_{\text{mean}}, \epsilon_{\text{norm}}, \epsilon_{\text{gap}}, \epsilon_{\text{dist}} \in (0,1)$.

1: Find $u_{\max} = \max_{s' \in \mathcal{S}} u(s')$, $s_{\max} = \arg\max_{s' \in \mathcal{S}} u(s')$ and $u_{\min} = \min_{s' \in \mathcal{S}} u(s')$ with success probability $1 - \frac{\delta}{4}$ using Fact 1.

2: Set $p(s_{\max}) = \hat{p}(s_{\max}|s,a) + \frac{d(s,a)}{2}$ and set $p(s') = \hat{p}(s'|s,a)$ for all $s' \in \mathcal{S} \backslash \{s_{\max}\}$.

3: Set $t = 1$.

4: Set $\text{high}^{(1)} = u_{\max}, \text{low}^{(1)} = u_{\min}$.

5: **while** $\left| \tilde{\Gamma}_{\leq c} - \frac{d(s,a)}{2} \right| > \epsilon_{\text{stop}}$ and $\left| c^{(t+1)} - c^{(t)} \right| > \frac{\epsilon_{\text{dist}}}{2}$ and $\left| \text{high}^{(t)} - \text{low}^{(t)} \right| > \epsilon_{\text{dist}}$ **do**

6:     Set $c^{(t)} = \text{low}^{(t)} + \left( \frac{\text{high}^{(t)} - \text{low}^{(t)}}{2} \right)$.

7:     Compute the estimate $\tilde{\Gamma}_{\leq c^{(t)}}$ of $\sum_{s':p(s') \leq c^{(t)}} \hat{p}(s'|s,a)$ using Lemma 1 with additive error $\epsilon_{\text{norm}}$ and success probability $1 - \frac{\delta}{4 \log S}$.

8:     **if** $\tilde{\Gamma}_{\leq c^{(t)}} \geq \frac{d(s,a)}{2} + \epsilon_{\text{gap}}$ **then**

9:         Set $\text{high}^{(t+1)} = c^{(t)}, \text{low}^{(t+1)} = \text{low}^{(t)}$.

10:         $t = t + 1$.

11:     **else if** $\tilde{\Gamma}_{\leq c} \leq \frac{d(s,a)}{2} - \epsilon_{\text{gap}}$ **then**

12:         Set $\text{low}^{(t+1)} = c^{(t)}, \text{high}^{(t+1)} = \text{high}^{(t)}$.

13:         $t = t + 1$.

14:     **else**

15:         Return $c^{(t)}$.

16:     **end if**

17: **end while**

18: Find the state $\bar{s} = \arg\min_{s' \in \mathcal{S}} \{u(s') \geq c^{(t)}\}$ using Fact 3. If there exist $i, j \in [S]$ where $i \neq j$ such that $u(s_i) = u(s_j)$, then either one of them will be returned.

19: Compute the estimates $\tilde{\mu}, \tilde{\mu}_{\leq c^{(t)}}$ such that

$$\left| \tilde{\mu} - \sum_{s' \in \mathcal{S}} \hat{p}(s'|s,a) \cdot u(s') \right| \leq \epsilon_{\text{mean}}$$

$$\left| \tilde{\mu}_{\leq c^{(t)}} - \sum_{s':(s') \leq c^{(t)}} \hat{p}(s'|s,a) \cdot u_i(s') \right| \leq \epsilon'_{\text{mean}},$$

each with success probability $1 - \frac{\delta}{4}$ using Lemma 2.

20: Set $p(\bar{s}) = \tilde{\Gamma}_{\leq c} - \frac{d(s,a)}{2}$.

**Output:** $\tilde{\mu}_{\max} = \tilde{\mu} - \tilde{\mu}_{\leq c^{(t)}} + \frac{d(s,a)}{2} \cdot u_{\max} + p(\bar{s}) \cdot u(\bar{s})$.

---

*(Case 1)*: $\tilde{\Gamma}_{\leq c^{(t)}} \geq \frac{d(s,a)}{2} + \epsilon_{\text{stop}}$

$$\left| c^{(t+1)} - c^{(t)} \right| \leq \frac{\epsilon_{\text{dist}}}{2}$$

$$\Leftrightarrow \left| \text{low}^{(t+1)} + \frac{\text{high}^{(t+1)} - \text{low}^{(t+1)}}{2} - c^{(t)} \right| \leq \frac{\epsilon_{\text{dist}}}{2}$$

$$\Leftrightarrow \left| \text{low}^{(t)} - \frac{c^{(t)} - \text{low}^{(t)}}{2} - c^{(t)} \right| \leq \frac{\epsilon_{\text{dist}}}{2}$$

$$\Leftrightarrow \left| \frac{\text{low}^{(t)}}{2} - \frac{c^{(t)}}{2} \right| \leq \frac{\epsilon_{\text{dist}}}{2}$$

$$\Leftrightarrow \left| \frac{\text{low}^{(t)}}{2} - \frac{\text{high}^{(t)}}{2} \right| \leq \frac{\epsilon_{\text{dist}}}{2}$$

$$\Leftrightarrow \left| \text{high}^{(t)} - \text{low}^{(t)} \right| \leq \epsilon_{\text{dist}}.$$

(*Case 2*): $\tilde{\Gamma}_{\leq c^{(t)}} \leq \frac{d(s,a)}{2} - \epsilon_{\text{stop}}$

$$\left| c^{(t+1)} - c^{(t)} \right| \leq \frac{\epsilon_{\text{dist}}}{2}$$

$$\Leftrightarrow \left| \text{low}^{(t+1)} + \frac{\text{high}^{(t+1)} - \text{low}^{(t+1)}}{2} - c^{(t)} \right| \leq \frac{\epsilon_{\text{dist}}}{2}$$

$$\Leftrightarrow \left| c^{(t)} + \frac{\text{high}^{(t)} - c^{(t)}}{2} - c^{(t)} \right| \leq \frac{\epsilon_{\text{stop}}}{2}$$

$$\Leftrightarrow \left| \frac{\text{high}^{(t)} - c^{(t)}}{2} \right| \leq \frac{\epsilon_{\text{dist}}}{2}$$

$$\Leftrightarrow \left| \frac{\text{high}^{(t)} - \text{low}^{(t)}}{2} \right| \leq \frac{\epsilon_{\text{dist}}}{2}$$

$$\Leftrightarrow \left| \text{high}^{(t)} - \text{low}^{(t)} \right| \leq \epsilon_{\text{dist}}.$$

Now, we prove the correctness of Algorithm 2. After exiting the while loop, a cut-off point $c^{(t)}$ is obtained. We denote the cut-off point as $c$ for brevity. Then, we can bound

$$|\tilde{\mu}_{\max}(s,a) - \mu_{\max}(s,a)|$$

$$= \left| \tilde{\mu} - \tilde{\mu}_{\leq c} + \frac{d(s,a)}{2} \cdot u_{\max} + p(\bar{s}) \cdot u(\bar{s}) - \mu_{\max}(s,a) \right|$$

$$\leq \left| \tilde{\mu} - \tilde{\mu}_{\leq c} + \frac{d(s,a)}{2} \cdot u_{\max} - \left( \sum_{s' \in \mathcal{S}} \hat{p}(s'|s,a) \cdot u(s') - \sum_{s' \in \mathcal{S}: u(s') \leq c} \hat{p}(s'|s,a) \cdot u(s') + \frac{d(s,a)}{2} \cdot u_{\max} + p(\bar{s}) \cdot u(\bar{s}) \right) \right|$$

$$\leq \left| \tilde{\mu} - \sum_{s' \in \mathcal{S}} \hat{p}(s'|s,a) \cdot u(s') - \tilde{\mu}_{\leq c} + \sum_{s' \in \mathcal{S}: u(s') \leq c} \hat{p}(s'|s,a) \cdot u(s') \right|$$

$$\leq \left| \tilde{\mu} - \sum_{s' \in \mathcal{S}} \hat{p}(s'|s,a) \cdot u(s') \right| + \left| \tilde{\mu}_{\leq c} - \sum_{s' \in \mathcal{S}: u(s') \leq c} \hat{p}(s'|s,a) \cdot u(s') \right|$$

$$\leq \epsilon_{\text{mean}} + \epsilon'_{\text{mean}}.$$

Setting $\epsilon_{\text{mean}} = \epsilon'_{\text{mean}} = \frac{\epsilon}{2}$ yields the desired bound. A union bound of all steps in the algorithm succeeding leads to the state total success probability.

For the time complexity, finding $u_{\max}, u_{\min}, s_{\max}$ takes $O(\sqrt{S} \log \frac{1}{\delta})$ time by quantum minimum finding. At every iteration of the binary search, we use quantum norm estimation to approximately compute norm of the transition probability vector on entries that correspond to the good states, which takes $O\left( \frac{\sqrt{S}}{\epsilon_{\text{norm}}} \log \frac{1}{\delta} \right)$ time. Furthermore, it is known that binary search finds a target solution after $O(\log S)$ iterations. Considering that an additive error of $\epsilon_{\text{norm}}$ is incurred at the end of every iteration of binary search, the run time of binary search suffers an extra $O\left( \log \frac{1}{\epsilon_{\text{norm}}} \right)$ overhead. The desired $\tilde{c}$ is obtained after $\tilde{O}(\log S)$ iterations of binary search, after which we perform quantum mean estimation on the good states with respect to $\tilde{c}$. This takes time $\tilde{O}\left( \frac{1}{\epsilon_{\text{mean}}} \log \frac{1}{\delta} \right)$. In total, the run time of Algorithm 2 is

$$\tilde{O}\left( \sqrt{S} \left( \frac{1}{\epsilon_{\text{norm}}} + \frac{1}{\epsilon_{\text{mean}}} \right) \log \frac{1}{\delta} \right).$$

---

**Algorithm 3** Quantum extended value iteration

---

**Input:** Quantum access to estimates $\hat{\mathbf{p}}(\cdot|s,a)$ stored in $\mathsf{KP}_{\hat{\mathbf{p}}}$ and $\mathsf{KP}_{\mathbf{u}}$, distance $d(s,a)$ for a state-action pair $(s,a)$, failure probability $\delta \in (0,1)$, $u \in \mathbb{R}^S$, error $\epsilon \in (0,1)$.

1: Set $i = 0$.
2: Initialize $u^{(0)}(s) = 0$ for all $s \in \mathcal{S}$.
3: **for all** $s \in \mathcal{S}$ **do**
4:    Let $q^{(i+1)}(s,a) = \hat{r}(s,a) + d(s,a) + \tilde{\mu}_{\max}(s,a)$, where $\tilde{\mu}_{\max}(s,a)$ is evaluated by running Algorithm 2 with additive error $\frac{\epsilon}{2}$ and success probability $1 - \frac{\delta \pi^2}{48 S (i+1)^2}$ using Lemma 3.
5:    $\tilde{u}^{(i+1)}(s) \leftarrow$ Obtain $\max_{a \in \mathcal{A}} \left\{ q^{(i+1)}(s,a) \right\}$ with additive error $\epsilon$ and success probability $1 - \frac{\delta \pi^2}{48 S (i+1)^2}$ using Fact 2.
6: **end for**
7: Update $\mathsf{KP}_{\mathbf{u}}$.
8: Find $u_{\max}^{(i+1)}$ and $u_{\min}^{(i+1)}$ using Fact 1 with success probability $1 - \frac{\delta \pi^2}{24}$.
9: **while** $\max_{s \in \mathcal{S}} \left\{ \tilde{u}^{(i+1)}(s) - \tilde{u}^{(i)}(s) \right\} - \min_{s \in \mathcal{S}} \left\{ \tilde{u}^{(i+1)}(s) - \tilde{u}^{(i)}(s) \right\} > \epsilon$ **do**
10:    Set $i = i + 1$.
11:    Repeat Lines 3-8.
12: **end while**
13: **for do**all $s \in \mathcal{S}$
14:    Find $\tilde{\pi}^{(i)}(s) = \left\{ a \in \mathcal{A} : q(s,a) \geq \max_{a \in \mathcal{A}} q(s,a) - \epsilon \right\}$.
15: **end for**
**Output:** $\tilde{u}^{(i+1)}, \tilde{\pi}$.

---

Setting $\epsilon_{\text{mean}} = \epsilon_{\text{norm}} = \frac{\epsilon}{2}$ yields a total run time of $\tilde{O}\left( \frac{\sqrt{S}}{\epsilon} \log \frac{1}{\delta} \right)$.     □

Now, we propose the quantum extended value iteration algorithm. At every iteration, this algorithm uses Algorithm 2 as a subroutine to compute the inner maximization of Eq. (11). It then uses a generalization of minimum finding to obtain the value function for every state $s \in \mathcal{S}$. The steps of the quantum extended value iteration algorithm are detailed in Algorithm 3.

The lemma below states the guarantees of one iteration of quantum extended value iteration.

**Lemma 4** (Guarantees of one iteration of quantum extended value iteration). *Let $\epsilon, \delta \in (0,1)$. Fix $i \in \mathbb{Z}_{\geq 0}$. Given access to estimated rewards $\hat{r}(s,a)$, estimated maximum mean value $\tilde{\mu}_{max}(s,a)$ and distance $d(s,a)$ for a state-action pair, there exists a quantum algorithm that outputs an estimate $\tilde{u}^{(i+1)}(s)$ of the solution $u^{(i+1)}(s)$ to Eq. (11) such that $\tilde{u}^{(i+1)}(s) \geq u^{(i+1)}(s) - \epsilon$ with success probability at least $1 - \delta$ for all $s \in \mathcal{S}$. This requires $\tilde{O}\left( \frac{S^{1.5}\sqrt{A}}{\epsilon} \log \frac{1}{\delta} \right)$ time.*

*Proof.* We first analyze the correctness of Algorithm 3 for every $s \in \mathcal{S}$. By Lemma 3, Algorithm 2 returns $\tilde{\mu}_{\max}(s,a)$ such that

$$|\tilde{\mu}_{\max}(s,a) - \mu_{\max}(s,a)| \leq \frac{\epsilon}{2}.$$

Then by Fact 2, we get an estimate $\tilde{u}(s)$ such that

$$\tilde{u}(s) \geq u(s) - \epsilon,$$

where $u(s)$ is defined as in Eq. (5). A union bound of all steps in the algorithm succeeding leads to a total success probability of $1 - \delta$.

Now, we analyze the time complexity of the algorithm. For every $s \in \mathcal{S}$, we find the maximum of $q(s,a)$ over all $a \in \mathcal{A}$ in Algorithm 3. This takes $\tilde{O}\left( \frac{\sqrt{A}}{\epsilon} \log \frac{1}{\delta} \right)$ time. For every run of the maximum finding in Line 5, we run Algorithm 2 to find $\tilde{\mu}_{\max}(s,a)$ in $\tilde{O}\left( \frac{\sqrt{S}}{\epsilon} \log \frac{1}{\delta} \right)$ time. The run time till Line 6 is therefore $O\left( S^{1.5}\sqrt{A} \log \frac{1}{\delta} \right)$. Finding $u_{\max}$ and $u_{\min}$ takes $O(\sqrt{S} \log \frac{1}{\delta})$

time. Therefore, the total amount of time for a single run of quantum extended value iteration is $\tilde{O}\left(\frac{S^{1.5}\sqrt{A}}{\epsilon}\log\frac{1}{\delta}\right)$. $\qquad\square$

Before proceeding to proving convergence for quantum extended value iteration, we show that the policy chosen by the algorithm is always a policy with aperiodic transition matrix. Ref. [19] argued that extended value iteration always chooses a policy with aperiodic transition matrix. In particular, define set $E$ and $F$ as follows

$$E = \{\pi \in \Pi : \mathbf{P}_\pi \boldsymbol{\rho}^* = \boldsymbol{\rho}^*\}, \quad F = \{\pi \in \Pi : \mathbf{P}_\pi \text{ is aperiodic}\}. \tag{20}$$

Then, there exists an $i_0$ such that for all $i \geq i_0$,

$$\max_{\pi \in \Pi}\left\{\mathbf{r}_\pi + \mathbf{P}_\pi \mathbf{u}^{(i)}\right\} = \max_{\pi \in E \cap F}\left\{\mathbf{r}_\pi + \mathbf{P}_\pi \mathbf{u}^{(i)}\right\}.$$

Since quantum extended value iteration is erroneous, we replace the set $E$ by

$$E' = \{\pi \in \Pi : \|P_\pi \rho^* - \rho^*\|_\infty \leq \epsilon\} \tag{21}$$

and use the same argument as [19] to show that the same policy choice holds.

**Lemma 6.** *Let $\Pi$ be the set of all policies and let $\tilde{\boldsymbol{\mu}}_\pi$ be defined as in Eq. (6). Let $\{\tilde{\mathbf{u}}^{(i)}\}$ be a sequence generated by Algorithm 3 and let $E', F$ be defined as in Eqs. (20) and (21). Then, there exists an $i_0 \in \mathbb{Z}_+$ such that for all $i \geq i_0$,*

$$\max_{\pi \in \Pi}\{\mathbf{r}_\pi + \tilde{\boldsymbol{\mu}}_\pi^{(i)}\} = \max_{\pi \in E' \cap F}\{\mathbf{r}_\pi + \tilde{\boldsymbol{\mu}}_\pi^{(i)}\}.$$

*Proof.* Since there are only finitely many deterministic policies with aperiodic transition probabilities [1], there exists an $i_0$ and a set $\Pi'$ such that for $i \geq i_0$, $\arg\max_{\pi \in \Pi}\left\{\mathbf{r}_\pi + \tilde{\boldsymbol{\mu}}_\pi^{(i)} - \epsilon\mathbf{e}\right\} \in \Pi'$. Choose a $\pi' \in \Pi'$. Then, there exists a subsequence $\left\{\tilde{\mathbf{u}}^{(i_k)}\right\}$ such that

$$\tilde{\mathbf{u}}^{(i_k+1)} = \mathbf{r}_{\pi'} + \tilde{\boldsymbol{\mu}}_{\pi'}^{(i)} - \epsilon \geq \mathbf{u}^{(i_k+1)} - (i_k+1)\epsilon\mathbf{e}$$

by Claim 1. Dividing both sides of the equality by $i_k + 1$ and letting $k \to \infty$, we get

$$\boldsymbol{\rho}^* \geq \limsup_{k\to\infty}\frac{\tilde{\mathbf{u}}^{(i_k+1)}}{i_k+1} \geq \lim_{k\to\infty}\frac{\tilde{\mathbf{u}}^{(i_k+1)}}{i_k+1} \geq \lim_{k\to\infty}\frac{\mathbf{u}^{(i_k+1)} - (i_k+1)\epsilon\mathbf{e}}{i_k+1} = \boldsymbol{\rho}^* - \epsilon\mathbf{e} = \mathbf{P}_{\pi'}\rho^* - \epsilon\mathbf{e}$$

where the first inequality is due to Theorem 1, the second last equality follows from Fact 7 and the last inequality follows from the implication of Fact 7. On the other hand,

$$\tilde{\mathbf{u}}^{(i_k+1)} = \mathbf{r}_\pi + \tilde{\boldsymbol{\mu}}_\pi^{(i)} - \epsilon \leq \mathbf{u}^{(i_k+1)}$$

by Claim 1. Dividing both sides of the equality by $i_k + 1$ and letting $k \to \infty$, we get

$$\boldsymbol{\rho}^* - \epsilon\mathbf{e} \leq \liminf_{k\to\infty}\frac{\tilde{\mathbf{u}}^{(i_k+1)}}{i_k+1} \leq \lim_{k\to\infty}\frac{\tilde{\mathbf{u}}^{(i_k+1)}}{i_k+1} \leq \limsup_{k\to\infty}\frac{\tilde{\mathbf{u}}^{(i_k+1)}}{i_k+1} \leq \lim_{k\to\infty}\frac{\mathbf{r}_{\pi'} + \mathbf{P}_{\pi'}u^{(i_k)}}{i_k+1} = \mathbf{P}_{\pi'}\rho^*.$$

Therefore, $\pi' \in E' \cap F$. $\qquad\square$

Next, we define a $J$-stage span contraction as follows.

**Definition 5** ($J$-stage span contraction). *Let $0 \leq \mu < 1$ and $\mathbf{u}, \mathbf{v} \in \mathcal{V}$. Denote the span of a vector $v$ as*

$$sp(\mathbf{v}) = \max_{s\in\mathcal{S}}\{v(s)\} - \min_{s\in\mathcal{S}}\{v(s)\}.$$

*For some positive integer $J$, we say that an operator $L : \mathcal{V} \to \mathcal{V}$ is a $J$-stage span contraction if $L$ satisfies*

$$sp\left(L^J\mathbf{u} - L^Jv\right) \leq \nu sp\left(\mathbf{u} - \mathbf{v}\right).$$

In the following lemma, we show that Algorithm 3 will eventually terminate.

**Lemma 7.** *Let $\epsilon, \epsilon' \in (0,1)$. There exists some positive integer $k$ such that $\mathcal{L}'$ satisfies*

$$\max_{s \in \mathcal{S}} \left\{ \tilde{u}^{(k+1)}(s) - \tilde{u}^{(k)}(s) \right\} - \min_{s \in \mathcal{S}} \left\{ \tilde{u}^{(k+1)}(s) - \tilde{u}^{(k)}(s) \right\} \le \sigma,$$

*where $\sigma = \epsilon' + 2iJ\epsilon$.*

*Proof.* First, notice that by Claim 1,

$$\mathcal{L}'v(s) - \mathcal{L}'u(s) \le \mathcal{L}v(s) - \mathcal{L}u(s) + \epsilon$$

for all $s \in \mathcal{S}$. Taking the maximum on both sides gives

$$\max_{s \in \mathcal{S}} \left\{ \mathcal{L}'v(s) - \mathcal{L}'u(s) \right\} \le \max_{s \in \mathcal{S}} \left\{ \mathcal{L}v(s) - \mathcal{L}u(s) \right\} + \epsilon. \tag{22}$$

Again by Claim 1, we have

$$\mathcal{L}'v(s) - \mathcal{L}'u(s) \ge \mathcal{L}v(s) - \epsilon - \mathcal{L}u(s)$$

for all $s \in \mathcal{S}$. Taking the minimum on both sides gives

$$\min_{s \in \mathcal{S}} \left\{ \mathcal{L}'v(s) - \mathcal{L}'u(s) \right\} \ge \min_{s \in \mathcal{S}} \left\{ \mathcal{L}v(s) - \mathcal{L}u(s) \right\} - \epsilon. \tag{23}$$

Combining Eqs. (22) and (23), we have for some positive integer $J$,

$$\begin{aligned}
&sp\left( \mathcal{L}'\mathbf{u}^{(J)} - \mathcal{L}'\mathbf{v}^{(J)} \right) \\
&= \max_{s \in \mathcal{S}} \left\{ \mathcal{L}'u^{(J)}(s) - \mathcal{L}'v^{(J)}(s) \right\} - \min_{s \in \mathcal{S}} \left\{ \mathcal{L}'u^{(J)}(s) - \mathcal{L}'v^{(J)}(s) \right\} \\
&\le \max_{s \in \mathcal{S}} \left\{ \mathcal{L}v^{(J)}(s) - \mathcal{L}u^{(J)}(s) \right\} - \min_{s \in \mathcal{S}} \left\{ \mathcal{L}v^{(J)}(s) - \mathcal{L}u^{(J)}(s) \right\} + 2J\epsilon \\
&= sp\left( \mathcal{L}\mathbf{v}^{(J)} - \mathcal{L}\mathbf{u}^{(J)} \right) + 2J\epsilon \\
&= sp\left( \mathcal{L}^J\mathbf{u} - \mathcal{L}^J\mathbf{v} \right) + 2J\epsilon \\
&\le \nu sp\left( \mathbf{u} - \mathbf{v} \right) + 2J\epsilon,
\end{aligned}$$

where $0 \le \nu < 1$ and the last inequality is due to the fact that $\mathcal{L}$ is a $J$-stage span contraction [1]. By setting $\mathbf{v} = \mathbf{u}^{(0)}$ and $\mathbf{u} = \mathcal{L}\mathbf{u}^{(0)}$, we get

$$sp\left( \tilde{\mathbf{u}}^{(iJ+1)} - \tilde{\mathbf{u}}^{(iJ)} \right) \le \nu^i sp\left( \mathbf{u}^{(1)} - \mathbf{u}^{(0)} \right) + 2iJ\epsilon \le \epsilon' + 2iJ\epsilon,$$

where the last inequality is due to [1, Theorem 8.5.2(b)]. Setting $\sigma = \epsilon' + 2iJ\epsilon$ completes the proof. $\square$

Now, we are ready to prove the convergence of Algorithm 3.

**Theorem 3** (Convergence of quantum extended value iteration). *Let $\epsilon, \delta \in (0,1)$. Let $\mathcal{M}$ be the set of all MDPs with state space $\mathcal{S}$, action space $\mathcal{A}$, transition probabilities $\tilde{\mathbf{p}}(\cdot|s,a)$, and mean rewards $\tilde{r}(s,a)$ that satisfy Eq.(9) and (10) for given probability distributions $\hat{\mathbf{p}}(\cdot|s,a)$, values $\hat{r}(s,a) \in [0,1], d(s,a) > 0$, and $d'(s,a) \ge 0$. If $\mathcal{M}$ contains at least one communicating MDP, Algorithm 3 satisfies*

$$\boldsymbol{\rho}^* - \epsilon\mathbf{e} \le \lim_{i \to \infty} \frac{\tilde{\mathbf{u}}^{(i)}}{i} \le \boldsymbol{\rho}^*.$$

*Furthermore, terminating Algorithm 3 when*

$$\max_{s \in \mathcal{S}} \left\{ \tilde{u}^{(i+1)}(s) - \tilde{u}^{(i)}(s) \right\} - \min_{s \in \mathcal{S}} \left\{ \tilde{u}^{(i+1)}(s) - \tilde{u}^{(i)}(s) \right\} \le \epsilon,$$

*the greedy policy with respect to $\tilde{\mathbf{u}}^{(i)}$ is $\epsilon$-optimal.*

*Proof.* By Lemma 6, the optimal policy $\pi^*$ has aperiodic transition matrix. Replacing $E$ with $E' \cap F$, $\mathbf{u}^{(i)}$ with $\tilde{\mathbf{u}}^{(i)}$ and using Lemma 6 instead of Lemma 9.4.3, the proof of [1, Theorem 9.4.4] follows by the aperiodicity of $\mathbf{P}_\pi$ from Lemma 6. By Theorem 1, we showed that the value of $\frac{\tilde{\mathbf{u}}^{(i)}}{i}$ is bounded between $\boldsymbol{\rho}^* - \epsilon \mathbf{e}$ and h$o$ as $i \to \infty$.

Now, we prove the error bound. Define

$$\rho' = \frac{1}{2} \left[ \max_{s \in \mathcal{S}} \left\{ \tilde{u}^{(i+1)}(s) - \tilde{u}^{(i)}(s) \right\} + \min_{s \in \mathcal{S}} \left\{ \tilde{u}^{(i+1)}(s) - \tilde{u}^{(i)}(s) \right\} \right]$$

By the same approach as [1], observe that if $a \le b \le c$ for $a, b, c \in \mathbb{R}$ and $c - a < \epsilon$, then

$$\frac{\epsilon}{2} < \frac{a - c}{2} \le b - \frac{a + c}{2} \le \frac{c - a}{2} < \frac{\epsilon}{2}.$$

By Theorem 2, setting $\tilde{\mathbf{u}} = \tilde{\mathbf{u}}^{(i)}$, we get

$$|\rho' - \rho^*| \le \frac{\epsilon}{2}, \quad \left| \rho' - \rho^{\pi^\infty} \right| \le \frac{\epsilon}{2}.$$

By triangle inequality,

$$\left| \rho^{\pi^\infty} - \rho^* \right| = \left| \rho^{\pi^\infty} - \rho' + \rho' - \rho^* \right| \le |\rho' - \rho^*| + \left| \rho' - \rho^{\pi^\infty} \right| \le \epsilon. \qquad \square$$

## D  QUANTUM-ACCESSIBLE ENVIRONMENTS

In order to access the MDPs, we assume access to a quantum sampling oracle for the transition probabilities, a quantum oracle for the rewards and a quantum policy evaluation oracle (see Definitions 2 to 4). Using these oracles, we describe a Classical Sampling via Quantum Access (CSQA) [21] procedure in Algorithm 4.

---

**Algorithm 4** Classical sampling via quantum access

---

**Input:** Policy $\pi$, time step $t$
1: Prepare $\tilde{\phi}^{(1)} := |x^{(1)}\rangle$.
2: **for** $t' = 1, 2, \cdots, t - 1$ **do**
3:     Query $\mathcal{O}_\mathcal{X}$ on $|\tilde{\phi}^{(t')}\rangle |\bar{0}\rangle$ to compute $|\phi^{(t')}\rangle := \mathcal{O}_\mathcal{X} |\tilde{\phi}^{(t')}\rangle |\bar{0}\rangle$.
4:     Query $\mathcal{O}_\pi$ on $|\phi^{(t')}\rangle |\bar{0}\rangle$ to compute $|\phi'^{(t')}\rangle := \mathcal{O}_\pi |\phi^{(t')}\rangle |\bar{0}\rangle$.
5:     Query $\mathcal{O}_p$ on $|\phi'^{(t')}\rangle |\bar{0}\rangle$ and collect the fourth register as $|\tilde{\phi}^{(t'+1)}\rangle$.
6: **end for**
7: Query $\mathcal{O}_\mathcal{X}$ on $|\tilde{\phi}^{(t)}\rangle |\bar{0}\rangle$ to compute $|\phi^{(t)}\rangle := \mathcal{O}_\mathcal{X} |\tilde{\phi}^{(t)}\rangle |\bar{0}\rangle$.
8: Measure the resulting state in the standard basis of $\mathcal{S}$.

---

**Lemma 5.** *Given a policy $\pi$ and an integer $t \in \mathbb{Z}_+$. Let $d_\pi^{(t)}$ be the probability distribution over states $s \in \mathcal{S}$ at step $t$ according to $\pi$. Suppose we have access to oracle $\mathcal{O}_\mathbf{p}$, then there exists a quantum algorithm that outputs a sample of $s \sim d_\pi^{(t)}$ in time $O(t)$.*

*Proof.* We slightly modify the CSQA algorithm by [21]. Starting with $|\tilde{\phi}^{(1)}\rangle = |x^{(1)}\rangle$, CSQA performs a discretization to produce $|\phi^{(t')}\rangle$, followed by a quantum evaluation of $\pi$ on $|\phi^{(t')}\rangle$ to produce $|\phi'^{(t')}\rangle$. Then, the algorithm queries $\mathcal{O}_p$ on $|\phi'^{(t')}\rangle$ and obtains the fourth register as $|\tilde{\phi}^{(t'+1)}\rangle$. If

$$|\tilde{\phi}^{(t')}\rangle = \int_x \sqrt{d_\pi^{(t')}(x)} |x\rangle,$$

then by Eq. (12), the fourth register of $\mathcal{O}_p |\phi'^{(t')}\rangle |0\rangle$ is

$$|\tilde{\phi}^{(t'+1)}\rangle = \int_x \sqrt{d_\pi^{(t'+1)}(x)} |x\rangle.$$

This can be seen from the fact that

$$|\phi'^{(t')}\rangle |0\rangle = \int_x \sqrt{d_\pi^{(t')}(x)} |x\rangle |s\rangle |a\rangle |0\rangle \xrightarrow{\mathcal{O}_{p^{(t)}}} \int_{x,x'} \sqrt{d_\pi^{(t')}(x)p(x'|s,a)} |x\rangle |s\rangle |a\rangle |x'\rangle .$$

Therefore, the fourth register is

$$\int_{x'} \sqrt{\int_x d_\pi^{(t')}(x)p(x'|s,a)} |x'\rangle = \int_{x'} \sqrt{d_\pi^{(t'+1)}(x')} |x'\rangle = |\tilde{\phi}^{(t'+1)}\rangle .$$

Querying $\mathcal{O}_\mathcal{X}$ on $|\tilde{\phi}^{(t'+1)}\rangle$ and measuring $|\tilde{\phi}^{(t)}\rangle$ gives a classical sample $s^{(t)} \sim d_\pi^{(t)}$ by induction. $\qquad \square$

# E  PROOF OF THEOREM 4

In this section, we prove the regret bound in Theorem 4.

## E.1  SPLITTING IN EPISODES

Let $n_k(s,a)$ denote the number of times action $a$ is chosen in episode $k$ when being in state represented by $s$. Let the regret in episode $k$ be

$$\Delta_k := \sum_{s \in \mathcal{S}} \sum_{a \in \mathcal{A}} n_k(s,a) (\rho^* - r(s,a)) . \tag{24}$$

As in Section 5.2.2 of [18] (cf. Section 5.1 of [22] and Section 4.1 of [19]), with probability at least $1 - \frac{\delta}{12T^{5/4}}$, the regret of Algorithm 1 is upper bounded by

$$\sqrt{\frac{5}{8}T \log\left(\frac{8T}{\delta}\right)} + \sum_k \Delta_k . \tag{25}$$

## E.2  FAILING CONFIDENCE INTERVAL

In this subsection, we consider the regret when the true MDP is not contained in the set of plausible MDPs. As mentioned in the previous section, the estimates $\hat{r}(x,a)$ and $\hat{p}_k^{\text{agg}}(x,a)$ are computed using their respective samples on the discretized state-action pair $(s,a)$.

**Rewards**  Using the algorithm in Fact 4, one can obtain an estimate $\hat{r}(x,a)$ of $\mathbb{E}[\hat{r}(x,a)]$ such that

$$|\hat{r}(x,a) - \mathbb{E}[\hat{r}(x,a)]| \leq \frac{\sqrt{SA}}{\max\{1, N_k(s,a)\}}$$

with success probability at least $1 - \frac{\delta}{24T^{5/4}}$ using $\tilde{O}(\max\{1, N_k(s,a)\})$ calls to $\mathcal{O}_r$. Combining with Eq. (3), we have for all $s \in \mathcal{S}, a \in \mathcal{A}$

$$|r(x,a) - \hat{r}(x,a)| \leq LS^{-\alpha} + \frac{\sqrt{SA}}{\max\{1, N_k(s,a)\}} \tag{26}$$

with success probability at least $1 - \frac{\delta}{24T^{5/4}}$.

**Transition probabilities**  Using the algorithm in Fact 4, one obtains an estimate $p'^{\text{agg}}(\cdot|x,a)$ of $\hat{p}^{\text{agg}}(\cdot|x,a)$ such that

$$\|\hat{p}^{\text{agg}}(\cdot|x,a) - \mathbb{E}[\hat{p}^{\text{agg}}(\cdot|x,a)]\|_1 \leq \frac{S}{N_k(s,a)}$$

with success probability at least $1 - \frac{\delta}{24T^{5/4}}$ [21] using $\tilde{O}(N_k(s,a))$ calls to $\mathcal{O}_{p^{(t)}}$. Combining with Eq. (4), we have for all $a \in \mathcal{A}$ and $I_j$ for $j \in [n]$,

$$\|p^{\text{agg}}(\cdot|x,a) - \hat{p}^{\text{agg}}(\cdot|x,a)\|_1 \leq LS^{-\alpha} + \frac{S}{N_k(s,a)} \tag{27}$$

with success probability at least $1 - \frac{\delta}{24T^{5/4}}$.

**Regret when confidence interval fail**  Ref. [19] gave a regret bond for the case when the true MDP is not contained in the set of plausible MDPs. They showed that

$$\sum_k \Delta_k \mathbb{1}_{M_k \notin \mathcal{M}_k} \leq \sqrt{T} \tag{28}$$

with probability at least $1 - \frac{\delta}{12T^{5/4}}$. This bound was also used by [18, 22]. In our case, the this regret bound holds with the same probability.

### E.3  REGRET IN EPISODES WITH $M \in \mathcal{M}_k$

We now analyze the regret when the true MDP $M$ lies in set of plausible MDPs. Note that by the $\frac{\epsilon}{\sqrt{T}}$-optimal choice of $\tilde{\pi}_k$, it holds that $\tilde{\rho}_k^* := \rho^* \left( \tilde{M}_k \right) \geq \rho^* - \frac{\epsilon}{\sqrt{T}}$. Therefore,

$$\rho^* - r(x, a) \leq (\tilde{\rho}_k^* - \tilde{r}_k(x, a)) + (\tilde{r}_k(x, a) - r(x, a)) + \frac{\epsilon}{\sqrt{T}},$$

By Eqs. (13), (24) and (26), we have

$$\Delta_k \leq \sum_x n_k(x, \tilde{\pi}_k(x))(\tilde{\rho}_k^* - \tilde{r}_k(x, \tilde{\pi}_k(x))) + 2LS^{-1}\tau_k + 2\sqrt{SA}\sum_{s \in \mathcal{S}}\sum_{a \in \mathcal{A}} \frac{n_k(s, a)}{N_k(s, a)} + \frac{\epsilon}{\sqrt{T}}\sum_x n_k(x, \tilde{\pi}(x)), \tag{29}$$

where we abuse the notation $n_k(x, \tilde{\pi}(x)) := n_k(s, a)$ for $s$ that represents $x$ and $\tau_k := t_{k+1} - t_k$ denotes the length of episode $k$.

**Dealing with the transition functions**  The term $\sum_x n_k(x, \tilde{\pi}(x))(\tilde{\rho}_k^* - \tilde{r}_k(x, \tilde{\pi}(x)))$ can be analyzed similar to Section 5.2.4 of [18] and Section 5.1 of [22]. Namely, let $\tilde{\lambda}_k := \lambda(\tilde{\pi}_k, \cdot)$ be the bias function of policy $\tilde{\pi}_k$ in the optimistic MDP $\tilde{M}_k$. By the Poisson equation,

$$\tilde{\rho}_k^* - \tilde{r}_k(x, \tilde{\pi}_k(x))$$

$$= \int_{\mathcal{X}} \tilde{p}_k(dx'|x, \tilde{\pi}_k(x)) \cdot \tilde{\lambda}_k(x') - \tilde{\lambda}_k(x)$$

$$= \int_{\mathcal{X}} p(dx'|x, \tilde{\pi}_k(x)) \cdot \tilde{\lambda}_k(x') - \tilde{\lambda}_k(x) + \int_{\mathcal{X}} (\tilde{p}_k(dx'|x, \tilde{\pi}_k(x)) - p(dx'|x, \tilde{\pi}_k(x))) \cdot \tilde{\lambda}_k(x'). \tag{30}$$

The last term in Eq. (30) can be bounded by

$$\tilde{\mathbf{p}}_k(\cdot|x, a) - \mathbf{p}(\cdot|x, a) = (\tilde{\mathbf{p}}_k(\cdot|x, a) - \hat{\mathbf{p}}_k(\cdot|x, a)) + (\hat{\mathbf{p}}_k(\cdot|x, a) - \mathbf{p}_k(\cdot|x, a))$$

$$\leq 2LS^{-\alpha} + 2\frac{S}{N_k(s, a)}$$

using Eqs. (14) and (27). This gives

$$\sum_x n_k(x, \tilde{\pi}_k(x)) \int (\tilde{p}_k(dx'|x, \tilde{\pi}_k(x)) - p(dx'|x, \tilde{\pi}_k(x))) \tilde{\lambda}_k(x')$$

$$\leq 2HS \sum_{s \in \mathcal{S}}\sum_{a \in \mathcal{A}} \frac{n_k(s, a)}{N_k(s, a)} + 2HLS^{-1}\tau_k. \tag{31}$$

For the first term in Eq. (30), the same result from Equation (29) of [18] and Equation(18) of [22] holds with probability at least $1 - \frac{\delta}{12T^{5/4}}$, i.e.

$$\sum_k \sum_x n_k(x, \tilde{\pi}_k(x)) \left( \int p(dx'|x, \tilde{\pi}_k(x)) \cdot \tilde{\lambda}_k(x') - \tilde{\lambda}(x) \right)$$

$$\leq H\sqrt{\frac{5}{2}T \log \frac{8T}{\delta}} + HSA \log \frac{8T}{nA}. \tag{32}$$

### E.4   TOTAL REGRET

As in Ref. [22, 18, 19], main regret term in the MDP comes from a sum over all confidence intervals in the visited state-action pairs. In order to bound this term, we rove the following lemma.

**Lemma 8.** *For any sequence of positive numbers* $z_1, \cdots, z_n$ *with* $0 \leq z_k \leq Z_{k-1} :=$
$\max\{1, \sum_{i=1}^{k-1} z_i\}$,

$$\sum_{k=1}^{n} \frac{z_k}{Z_{k-1}} \leq \frac{2}{\log 2} \log Z_n$$

*Proof.* By Lemma 2 of [18], we have

$$\sum_{k=1}^{n} \frac{z_k}{Z_{k-1}^{1-\alpha}} \leq \frac{Z_n^\alpha}{2^\alpha - 1}$$

for any $\alpha \in (0, 1]$. Also, notice that for $\alpha \in (0, 1]$,

$$\sum_{k=1}^{n} \frac{z_k}{Z_{k-1}} \leq \sum_{k=1}^{n} \frac{z_k}{Z_{k-1}^{1-\alpha}} \leq \frac{Z_n^\alpha}{2^\alpha - 1}.$$

It suffices to find the value of $\alpha$ that minimizes $\frac{Z_n^\alpha}{2^\alpha - 1}$. Taking the derivative of $\frac{Z_n^\alpha}{2^\alpha - 1}$ with respect to $\alpha$ and letting it be 0, we get

$$-Z_n^\alpha \left(-2^\alpha \log Z_n + \log Z_n + 2^\alpha \log 2\right) = 0$$
$$2^\alpha \log Z_n - 2^\alpha \log 2 = \log Z_n$$
$$2^\alpha \left(\log Z_n - \log 2\right) = \log Z_n$$
$$2^\alpha = \frac{\log Z_n}{\log Z_n - \log 2}$$
$$\alpha = \log_2 \left(\frac{\log Z_n}{\log Z_n - \log 2}\right).$$

Then,

$$\sum_{k=1}^{n} \frac{z_k}{Z_{k-1}} \leq \frac{Z_n^{\log_2\left(\frac{\log Z_n}{\log Z_n - \log 2}\right)}}{\frac{\log Z_n}{\log Z_n - \log 2} - 1} = \frac{(\log Z_n - \log 2) Z_n^{\log_2\left(\frac{\log Z_n}{\log Z_n - \log 2}\right)}}{\log 2} = \frac{\log(Z_n/2)}{\log 2} Z_n^{\log_2\left(\frac{\log Z_n}{\log(Z_n/2)}\right)}$$

$$= \frac{\log(Z_n/2)}{\log 2} \left(2^{\log_2 Z_n}\right)^{\log_2\left(\frac{\log Z_n}{\log Z_n - \log 2}\right)} = \frac{\log(Z_n/2)}{\log 2} \left(\frac{\log Z_n}{\log(Z_n/2)}\right)^{\log_2 Z_n}$$

$$= \frac{\log(Z_n/2)}{\log 2} \left(\frac{\log(Z_n/2) + \log 2}{\log(Z_n/2)}\right)^{\log_2 Z_n} = \frac{\log(Z_n/2)}{\log 2} \left(1 + \frac{\log 2}{\log(Z_n/2)}\right)^{\log_2 Z_n}$$

$$\leq \frac{\log(Z_n/2)}{\log 2} \left(10^{\frac{\log 2}{\log(Z_n/2)}}\right)^{\log_2 Z_n} = \frac{\log(Z_n/2)}{\log 2} \left(10^{\log 2}\right)^{\frac{\log z_n}{\log(Z_n/2)}} \leq \frac{2}{\log 2} \log Z_n.$$

where the first inequality uses the fact that $\log(1 + x) \leq x$ and the last inequality is due for the first inequality and the fact that $\log x$ is monotonically increasing for $x \in \mathbb{R}_+$. for the second inequality. $\qquad \square$

We note that a generalized version of Lemma 8 is given in Lemma 2 of [18]. However, the authors claimed that their lemma holds for all $\alpha \in [0, 1]$, which is not the case.

We now bound the total regret. Summing up $\Delta_k$ over all episodes with $M \in \mathcal{M}_k$, we obtain, by Eqs. (29) to (32),

$$\sum_k \Delta_k \mathbb{1}_{M \in \mathcal{M}_k}$$

$$\leq 2LS^{-1}\tau_k + 2\sqrt{nA}\sum_k\sum_{s\in\mathcal{S}}\sum_{a\in\mathcal{A}}\frac{n_k(s,a)}{N_k(s,a)} + H\sqrt{\frac{5}{2}T\log\frac{8T}{\delta}} + HSA\log\frac{8T}{SA}$$

$$+ 2HS\sum_k\sum_{s\in\mathcal{S}}\sum_{a\in\mathcal{A}}\frac{n_k(s,a)}{N_k(s,a)} + 2HLS^{-\alpha}\tau_k + \frac{\epsilon}{\sqrt{T}}\sum_k\sum_{s\in\mathcal{S}}\sum_{a\in\mathcal{A}}n_k(s,a) \tag{33}$$

Notice that by definition, $\tau_k \leq T$ and by Lemma 8, we have

$$\sum_k\sum_{s\in\mathcal{S}}\sum_{a\in\mathcal{A}}\frac{n_k(s,a)}{N_k(s,a)} \leq \frac{2}{\log 2}\sum_{s\in\mathcal{S}}\sum_{a\in\mathcal{A}}\log(N(s,a)) \leq \frac{2}{\log 2}\log(SAT)$$

due to Jensen's inequality, the definition $N(s,a) := \sum_k n_k(s,a)$ such that $\sum_{s\in\mathcal{S}}\sum_{a\in\mathcal{A}}N(s,a) = T$ [19]. Then, from Eq. (33), we have

$$\sum_k\Delta_k\mathbb{1}_{M\in\mathcal{M}_k} \leq 2LTS^{-\alpha} + \frac{4}{\log 2}\sqrt{SA}\log(SAT) + H\sqrt{\frac{5}{2}T\log\frac{8T}{\delta}} + HSA\log\frac{8T}{SA}$$

$$+ \frac{4}{\log 2}HS\log(SAT) + 2HTLS^{-\alpha} + \epsilon\sqrt{T}. \tag{34}$$

### E.5 TOTAL REGRET

By Eqs. (25), (28) and (34), we have

$$\sum_k\Delta_k = \sum_k\Delta_k\mathbb{1}_{M_k\notin\mathcal{M}_k} + \sum_k\Delta_k\mathbb{1}_{M_k\in\mathcal{M}_k} + \sqrt{\frac{5}{8}T\log\frac{8T}{\delta}}$$

$$\leq \sqrt{T} + 2LTS^{-\alpha} + \frac{4\sqrt{SA}}{\log 2}\log(SAT) + H\sqrt{\frac{5}{2}T\log\frac{8T}{\delta}} + HSA\log\frac{8T}{SA}$$

$$+ \frac{4}{\log 2}HS\log(SAT) + 2HTLS^{-\alpha} + \epsilon\sqrt{T} + +\sqrt{\frac{5}{8}T\log\frac{8T}{\delta}}$$

$$\leq 2(H+1)LTS^{-\alpha} + (14+15H)SA\log\frac{SAT}{\delta} + (2H+3)\sqrt{T}\log\frac{SAT}{\delta} \tag{35}$$

with probability at least $1 - \frac{\delta}{4T^{5/4}}$. Since $\sum_{T=2}^{\infty}\frac{\delta}{4T^{5/4}} < \delta$, a union bound over all possible values of $T$ gives

$$\sum_k\Delta_k \leq 2(H+1)LTS^{-\alpha} + (14+15H)SA\log\frac{SAT}{\delta} + (2H+3)\sqrt{T}\log\frac{SAT}{\delta}$$

with probability at least $1 - \delta$.

**Remark 1.** *The general $d$-dimensional case is almost similar the 1-dimensional case, with the only difference being that the discretization now has $n^d$ states. Replacing $S$ with $S^d$ and setting $S = T^{\frac{1}{1+2d\alpha}}$ bounds the regret by $\tilde{O}\left(T^{\frac{1}{1+d\alpha}}\right)$ when $d\alpha < 1$ and $\tilde{O}(\sqrt{T})$ when $d\alpha \geq 1$.*

