# OpenReview forum: "Quantum Algorithm for Online Learning of MDPs with Continuous State Space"
_ICLR.cc/2025/Conference — Submitted to ICLR 2025_

### Official Review · Reviewer_LU4a · 2024-10-31

**Soundness:** 3
**Presentation:** 3
**Contribution:** 3
**Rating:** 6
**Confidence:** 3

**Summary:**

Whether quantum algorithms could speedup machine learning tasks is of great interest in quantum computing.  In this paper, the authors gave a quantum algorithm for learning Markov Decision Processes (MDPs) with continuous state space in the average reward model. Their algorithm achieves a better regret bound in 1-dimensional state space and less complexity per iteration compared with formal classical algorithms. To achieve this, they proposed a quantum extended value iteration subroutine and proved approximate convergence guarantees for an approximate analog of value iteration.

**Strengths:**

- The authors gave a novel quantum algorithm that achieves a better regret bound in 1-dimensional state space and may have less complexity per iteration compared with formal classical algorithms, suggesting quantum advantages in solving this problem.
- They proposed a quantum extended value iteration subroutine and proved approximate convergence guarantees for an approximate analog of value iteration, theoretically analyzing their algorithm with provable guarantees.

**Weaknesses:**

- Althought the paper obtained a better regret bound $O(T^{1/2})$ for $1$-dimensional state space, it seems that this improvement does not hold for state space with higher dimensions (>3). Could the authors explain why this is the case, and explain possible limitations?
- The change of the iteration complexity is from $O(S^2 A)$ (classical) to $O(S^{1.5}\sqrt{A}/\varepsilon)$ (quantum), which is subquadratic.
- The techniques to obtain quantum speedups are quantum mean estimation and norm estimation, which looks not so novel. Could the author explain their technical novelties from a quantum computing perspective?

**Questions:**

- In line 113-114, they said "Moreover, (all or part of) the entries of $u$ can be classically updated by writing new values into the KP-tree in
 $\widetilde{O}(S)$ time." I think this could be a little misleading. For example, executing an "initialization" operation which set the vector stored in the KP tree to be a zero vector could be done in $\widetilde{O}(1)$ time. A more formal statement concerning this point could be referred to Theorem 2.11 in the arxiv version of [26].
- In line 229, "From hereon" -> "From here on".
- Line 828, "ancilla quantum sates" -> "ancillary quantum states"
- In line 843, "Let $W\subset S$ be the set of entries that satisfy some given condition". Could the authors explain more about the conditions for W?

---

### Official Review · Reviewer_eSTi · 2024-11-03

**Soundness:** 3
**Presentation:** 3
**Contribution:** 2
**Rating:** 6
**Confidence:** 3

**Summary:**

In this paper, the authors tackle the problem of online learning in MDPs with continuous state spaces under the average reward model. They propose a quantum algorithm that achieves a regret bound of $\tilde{O}(\sqrt{T})$ in one-dimensional continuous state spaces, representing a quantum speedup over the classical $\tilde{O}(T^{2/3})$ regret bound. Additionally, they establish a $\tilde{O}(T^{1-1/2d})$ upper bound for general d-dimensional state spaces. To accomplish these results, the authors introduce a quantum adaptation of the value iteration method, which provides a subquadratic speedup concerning the state space size and a quadratic speedup for the action space size. They also present a thorough theoretical analysis, demonstrating that the proposed quantum-extended value iteration converges to the optimal average reward within a small additive error.

**Strengths:**

The problem of online learning in continuous-state MDPs is both significant and broadly impactful, and this work introduces a novel quantum algorithm that achieves a quantum speedup on the number of time steps, as well as on the size of state space and action space.

This paper extends the value iteration subroutine into quantum domain, which might be useful in other quantum machine learning algorithms.

**Weaknesses:**

This paper currently includes minimal preliminaries on quantum computing, which may limit accessibility for readers outside the quantum computing field. A more comprehensive introduction on quantum computing, such as basic concepts of qubits and unitaries, the intuition on quantum speedup, and the rationale for quantum-accessible environments, would enhance comprehension and broaden the paper’s appeal to readers from diverse research backgrounds.

**Questions:**

Please consider the suggestion in the weaknesses section.

---

### Official Review · Reviewer_TaAy · 2024-11-05

**Soundness:** 3
**Presentation:** 2
**Contribution:** 2
**Rating:** 5
**Confidence:** 3

**Summary:**

This submission explores quantum reinforcement learning (RL) algorithms for Markov Decision Processes (MDPs) with continuous state spaces (referred to as Lipschitz-MDPs) under an infinite-horizon average-reward framework. The work extends classical reinforcement learning approaches in the same setting and claims an improved regret rate when compared to classical computational settings. Additionally, it introduces a quantum-enhanced version of value iteration, getting some computational speed up.

**Strengths:**

Developing a new reinforcement learning (RL) algorithm within the quantum computing framework holds promise for future advancements, and the improved regret is interesting.

**Weaknesses:**

I think the paper needs to be crystal clear about how the algorithms in the quantum computing setting are differentiated from the classical setting. The current writing does not sufficiently highlight what makes quantum computing interesting in terms of the algorithm design and analysis, and what is an additional layer (novel contribution) that the submission adds on in both settings.

-	The claim is that the regret rate is improved: it is not clear how this gain comes from the quantum computational setting.

-	I am not sure section 4.1 has anything to do with the specific constraints of quantum computing setting. I do not clearly see why the approximate value iteration is surprising (as it takes one page to describe that).

-	I wonder why the focus is specifically on the Lipschitz-MDP. Are there any other quantum results for Tabular/Linear MDPs? What are the benefits for more standard models?


Below are more detailed comments:

-	Line 71-12: Is the improvement in regret-guarantees a fair comparison to classical settings? Bounds given by [18] – does it match the classical lower bound, and quantum setting can break it?

-	Line 108-114: Discussion on KP-trees is so superficial. It doesn’t help at all for non-quantum experts like myself, and I do not think this level of discussion is not useful for quantum experts either.

-	Lemma 3: Why $\sqrt{S}/\epsilon$ is necessarily better than S^2A that without epsilon dependence?

-	Line 330: “only collect quantum states via quantum-accessible environments” – is this a good thing or bad thing? If it is a bad thing, how can the regret be improved?

-	Eq (12): what is garbage(x)?

-	Definition 4: why sqrt is over pi?

-	Algorithm 1: Could you clarify how that is different from UCRL2?

Overall, I am genuinely interested in what makes algorithms different in quantum computing settings, but the current writing of the paper obscures many important points on what makes “quantum” setting interesting, compared to classical settings. At this moment, I am inclined to rejection.

**Questions:**

See Weaknesses

---

> ### Comment · Area_Chair_Jf8e · 2024-11-27
>
> Dear Reviewer,
>
> The authors have provided their rebuttal to your comments/questions. Given that we are not far from the end of author-reviewer discussions, it will be very helpful if you can take a look at their rebuttal and provide any further comments. Even if you do not have further comments, please also confirm that you have read the rebuttal. Thanks!
>
> Best wishes,
> AC

---

### Meta-Review · Area_Chair_Jf8e · 2024-12-12

**Metareview:**

This paper proposed a quantum algorithm for online learning of MDPs with continuous state space that achieves a O(sqrt{T}) regret, which improves over the classical best-known results with regret O(T^{2/3}). The algorithm is based on the classical online UCCRL algorithms by Ortner and Ryabko (NeurIPS 2012). It also uses quantum extended value iteration as a subroutine. The result is purely theoretical.

The reviewers and the area chair found the technical result interesting, in particular it found a task in RL that algorithms on quantum computers can solve better than best-known classical algorithms. However, the paper also has notable weaknesses:

1) This work is a bit incremental compared to Ref. [21], which had achieved an exponential improvement in learning tabular and value-target MDPs. The style of the current manuscript is similar - the setting of continuous state space is new to quantum, but the improvement is only polynomial, not exponential as in that existing result. The algorithm also has similarity - Ref. [21] gave a quantum version of UCRL, and this paper gave a quantum version of UCCRL.

2) This work needs to clarify the motivation for studying continuous state space. As far as I know, in current RL research continuous state space MDP is not a specific topic with many studies. It would be better if the authors can motivate which practical problems in RL must use the description of continuous state space. In addition, it would be helpful to discuss more on whether the oracles in the continuous state space setting can be prepared. For instance, in Page 3 it assumes a query oracle $O_u$ that performs the mapping $O_u: |s⟩|0⟩ \to |s⟩|u(s)⟩$ for any $s \in S$ in time $O(\text{poly}(\text{log} S))$, but this is finite only when the state space is finite (and not continuous)? Which practical settings with continuous state space can this be efficient?

Considering these weaknesses, the decision is to reject this paper at ICLR 2025. The authors should also consider improving and clarifying over these points in future versions.

A typo in Line 496: rewardds -> rewards

**Additional Comments On Reviewer Discussion:**

The authors addressed the comments in initial review during the rebuttal. The reviewers read those but there weren't further discussions. Reviewers eSTi and TaAy did not change the score, while the reviewer LU4a changed the score from 5 to 6. Nevertheless, there is no support from the reviewers during the discussion period.

---

### Decision · Program_Chairs · 2025-01-22

Reject